

# Analysis of an intense O₃ pollution episode in the Atlantic Coast of the Iberian Peninsula using photochemical modeling: characterization of transport pathways and accumulation processes.

Eduardo Torre-Pascual[1], Gotzon Gangoiti[1], Ana Rodríguez-García[1], Estibaliz Sáez de Cámara[1], Joana Ferreira[2], Carla Gama[2], María Carmen Gómez[1], Iñaki Zuazo[1], Jose Antonio García[1], Maite de Blas[1].

[1]Faculty of Engineering Bilbao, University of the Basque Country (UPV/EHU), Bilbao, 48013, Spain

[2]CESAM & Department of Environment and Planning, University of Aveiro, Aveiro, 3810-193, Portugal

*Correspondence to:* Eduardo Torre-Pascual (eduardo.delatorre@ehu.eus)

**Abstract.** A tropospheric O₃ pollution episode over the Atlantic Coast of the Iberian Peninsula during August 2-6 in 2018 has been analyzed. The episode was characterized by a permanent wind shear throughout the entire period, making the observed ozone surface distribution especially difficult to explain. A new methodology is described analyzing upper-level atmospheric parameters, such as temperature, wind direction, wind speed, and O₃ concentrations, added to the traditional use of surface parameters, using WRF-CAMx models and available surface and upper-air observations. Results indicate that the episode was characterized by a first phase of a sudden increase in O₃ concentrations produced by fumigation and interregional transport processes within the Iberian Peninsula, followed by a continental O₃ transport from Europe to the Atlantic Coast. An Atlantic front produced the dissipation of the episode, generating an "ozone front" heading the cold front passage across the region.

## 1. Introduction

Southern European countries are heavily exposed to high tropospheric ozone (O₃) concentrations, particularly those surrounding the Mediterranean Basin (ETC/ACM, 2018; EEA, 2019). According to the last report on air quality in Europe in 2020 O₃ levels that year were lower than in previous years, but still high with maxima found in central Europe, some Mediterranean countries, and Portugal (EEA, 2022). Accumulation, transport, and recirculation processes behind these high concentrations have been extensively analyzed in the Western Mediterranean Basin and Eastern Iberia during the last 40 years (Millán et al., 1997; Gangoiti et al., 2001; Querol et al., 2016). However, O₃ episodes in the Atlantic Coast of the Iberian Peninsula (IP), specifically in Northern Atlantic Iberia (NAI) and Western Atlantic Iberia (WAI), have not been examined in detail. In this region, significant episodes of tropospheric O₃ with EC/50/2008 EU Directive exceedances have occurred.

In NAI, data from the summer flight campaigns of the European MECAPIP project (Millán et al., 1992, 1997) revealed long-range transport of photochemical pollutants from the English Channel into the Basque Country (BC) (Alonso et al., 2000). Pollutants are usually transported under the typical summer synoptic scenario, with the Azores High extending a ridge of high pressures over the Bay of Biscay and pushing northerly winds over NAI.

Gangoiti et al. (2002, 2006a) showed the importance of vertical layering and transport in the generation of intense O₃ episodes in the BC under a different synoptic situation, with persistent northeasterly winds associated with blocking anticyclones over the British Isles. That work also documented the importation of pollutants into the BC from several European source regions during the build-up of episodes, including the Iberian Peninsula. Valdenebro et al. (2011) showed how O₃ transport efficiency increased after the





formation of accumulation layers of polluted air masses aloft, which can travel large distances within a
stably stratified Maritime Boundary Layer (MBL) or over the stable nocturnal surface boundary layer over
land. The latter study demonstrated that transport from and to the Ebro and Douro valleys, both located in
the IP, plays a main role in $O_3$ episodes in the BC. Sáez de Cámara et al. (2018) documented that $O_3$
observations in background areas of the BC may have production and transport of local origin from
surrounding areas during midday, and a contribution from the arrival of polluted air masses in the afternoon
during the accumulation and peak phases.

Past studies for WAI in Portugal showed typical temporal patterns with maximum mean monthly
concentrations during spring, and maximum hourly concentrations during summer (Pires et al., 2012).
Concentrations are higher in inland and rural areas than in urban regions. However, $O_3$ episodes, with
concentrations above the thresholds defined for the protection of human health, also occur in urban regions.
Several studies (Evtyugina et al., 2007; Monteiro et al., 2012, 2016) showed the importance of sea breeze
circulation in the build-up of $O_3$ episodes through the Portuguese coast, pointing to the importance of
precursors emitted in coastal areas and $O_3$ production along the transport towards inland areas. Hertig et al.
(2020) showed that in Portugal the occurrence of $O_3$ and heat wave events had the strongest relationship
for eastern and northeastern inflow, highlighting the importance of the advection of $O_3$ pollution from the
continental parts of the Iberian Peninsula. In addition to the regular anthropogenic (e.g., traffic, industry,
energy production) and biogenic (natural) sources, extraordinary events such as forest fires play an
important role in the $O_3$ episodes registered in Portugal (Adame et al., 2012).

In this article, for the first time and as far as the authors know, the tropospheric $O_3$ problem is approached
in an integral way for the Atlantic Coast of the Iberian Peninsula. A 5-day pollution episode affecting two
countries, Spain and Portugal, has been analyzed. The episode described in this paper stands out for its
intense and simultaneous rise along NAI and WAI during August 2-6 in 2018 (see Section 3.2). We utilized
the modeling system described in Section 2 to examine the local-to-interregional transport and
accumulation of $O_3$ within and between two countries, as well as between these countries and the rest of
Europe. Valdenebro et al. (2011) hypothesized about the possibility of an $O_3$ and pollutants transport
pathway in the Atlantic axis of the IP. This hypothesis, together with the fact that Spain and Portugal share
three main air basins draining from central Iberia into the Atlantic, implies that the analysis should be
carried out as a whole for the two regions: Northern Atlantic Iberia (NAI) and Western Atlantic Iberia
(WAI), as shown in Figure 1. We expose how interregional transport of $O_3$ is a key element in explaining
the observed evolution of this episode.


## 1.1 Area Description

The Iberian Peninsula, with Spain and Portugal, has a complex topography with numerous mountain ranges,
with an average altitude among the highest in Europe. IP is surrounded by the Bay of Biscay to the North,
the Mediterranean Sea to the East, the Mediterranean and Atlantic Ocean to the South, where both meet
through the Strait of Gibraltar, and by the Atlantic Ocean to the West (Figure 1). It is separated from the
European continent by the Pyrenees Mountain range and a high central plateau largely occupies its surface.
The rivers flowing into the sea produce numerous air basins and valleys that are decisive for studying
atmospheric pollution due to their particular wind regimes.

To the North, parallel to the coast it is located the Cantabrian Range, with elevations of more than 2,500 m
ASL in its central zone. This mountain range extends from West to East to the western end of the Pyrenees
and separates the Northern coast of the IP from the Northern peninsular Plateau. Basque Country is in the
link between the Northern coast of IP and the Pyrenees, with lower altitude mountains, usually below 1,500
m ASL. To the West of IP, in Portugal, there are two different areas: north of the Tagus River where
mountainous areas also predominate, and south with flatter landscapes and a few low mountains. Most of
Portugal is below 400 m ASL, and the highest altitudes are in the Serra da Estrela, which forms a
continuation of the Spanish Central System. Both countries share the valleys of the Douro, Tagus, and
Guadiana rivers.



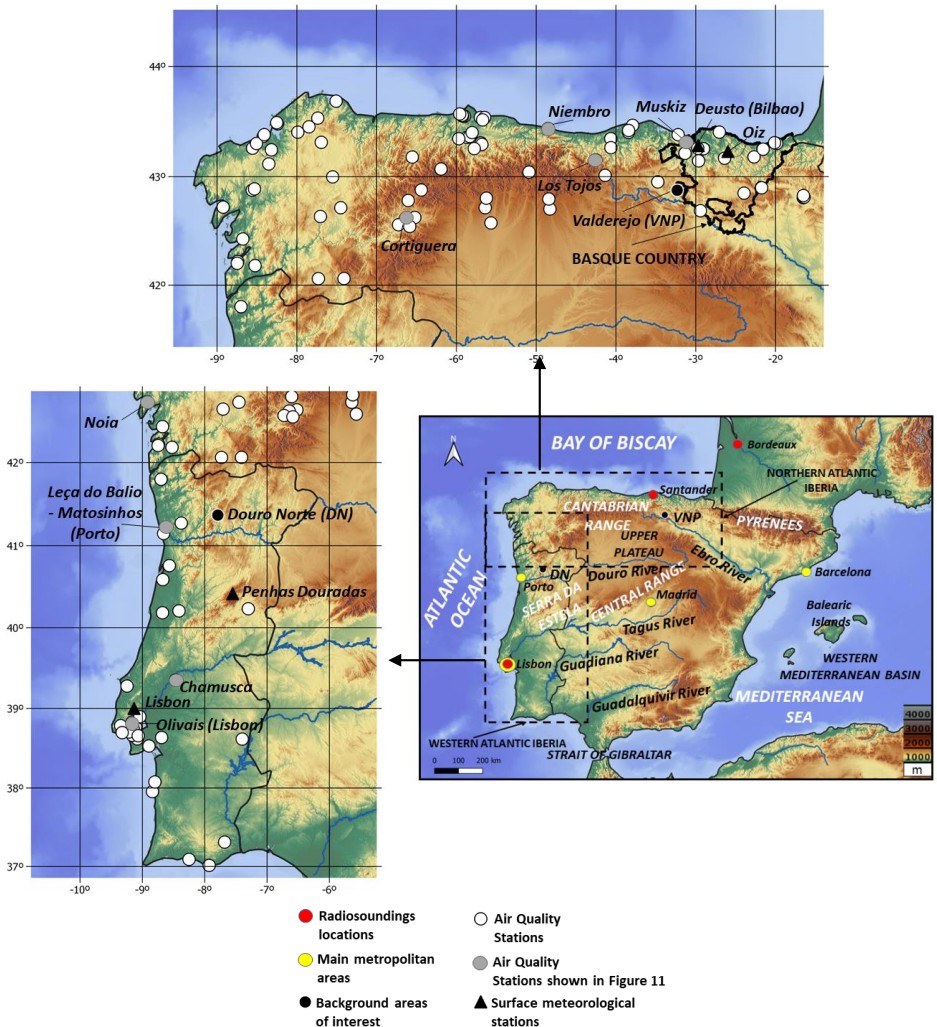

**Figure 1. Topographic map of the Iberian Peninsula (bottom-right): The territory delimited to the left towards the Atlantic Ocean is Portugal, to its right Spain, and to the north of the Pyrenees, France. Upper detailed map: North Atlantic Iberia (NAI) and left detailed map: Western Atlantic Iberia (WAI).**

Over the last years, the highest $O_3$ hourly averaged concentration registered in NAI and WAI have been continuously measured in rural mountainous areas such as Valderejo Natural Park (VNP) station, in the 100 Basque Country (Spain), and Douro Norte (DN) station, in Alvão Natural Park (Portugal). In both stations, $O_3$ exceedances are numerous, and $O_3$ levels are affected by primary pollutants (volatile organic compounds (VOCs) and nitrogen oxides (NOx)) emitted on their corresponding coastline, that are transported inland due to the sea breeze circulation. Besides the contribution of local sources, the concentration profiles reflect the influence of atmospheric transport on a synoptic or regional scale (Evtyugina et al., 2009; Carvalho et 105 al., 2010; Monteiro et al., 2012; Borrego et al., 2013, 2016; de Blas et al., 2019; Gómez et al., 2020). All these studies have analyzed various $O_3$ episodes for specific regions of the Iberian Atlantic Coast. However, there is a lack of a common modeling and assessing methodology for the whole region. The mechanisms producing $O_3$ episodes occurring simultaneously in the two sub-regions (NAI and WAI) of the Iberian Atlantic Coast are still in the process of being further documented.





**1.2. Objectives**

The main objective of this paper is to analyze a tropospheric $O_3$ episode with a remarkable intensity over a large region, covering Northern Atlantic Iberia (NAI) and Western Atlantic Iberia (WAI), and to establish possible $O_3$ interregional pathways between these regions and with the rest of the Iberian Peninsula and the European continent. For this purpose, we have established a methodology based on high-resolution

meteorological and photochemical modeling to analyze the surface concentration and vertical distribution of $O_3$. The presence of a permanent wind shear throughout the entire episode added special complexity and posed a challenge to the search for the origin of the observed $O_3$ impact and the selection of the most appropriate reduction policies.

This paper is organized as follows: Section 2 describes the methodology employed, containing 2

subsections. Section 2.1. refers to the modeling system used, and Section 2.2. the validation method. The results and discussion are presented in Section 3, divided into three subsections, analyzing the meteorology of the episode in 3.1, $O_3$ concentrations in 3.2, and statistical evaluation in 3.3. Finally, in Section 4 we detail the conclusions of this study.

**2.   Methodology**

In recent years Chemical Transport Models (CTM) have been used to simulate and analyze short-duration pollution episodes in IP (Valverde et al., 2016; Escudero et al., 2019; Pay et al., 2019). The use of fine grids in models (with high horizontal spatial resolutions of 1-3 km) has given good results in environments with complex topography, where mesoscale processes become particularly relevant for the interpretation of the

$O_3$ production, accumulation, transport, and decay (Jiménez et al., 2006; Monteiro et al., 2009). High horizontal spatial resolution is also especially recommended when describing $O_3$ variability in industrial and urban areas (Baldasano et al., 2011).

We have used a photochemical modeling system configuration, combining meteorological, emission, and photochemical simulations. Models' execution (for Initial and Boundary conditions, among others) and

validation require a variety of experimental data, all of them described throughout this section. Model results have been processed in order to analyze and represent vertical cross sections of the atmosphere, and we have calculated integrated $O_3$ concentrations from near-surface atmospheric levels up to 2,500 m Above Ground Level (AGL), according to the atmospheric thickness above the surface at which $O_3$ accumulates (Querol et al., 2018).

**2.1.   Simulations**
**2.1.1.   Meteorology**

The meteorological parameters required for air quality simulations were obtained using the Weather Research and Forecasting model (WRF), version 3.9.1.1 (Skamarock et al., 2008), using a modeling period from July 26 to August 9, 2018. We defined 3 domains with different resolutions (Table 1) and Lambert

Conformal projection, as shown in Figure 2, with the center of the coarser domain at 45°N and 2.5°W, and 50°N and 35°N as true latitudes for the projection. The first grid (d01) extension covers a large part of the European continent and Northern Africa with a 27 km horizontal resolution. This domain is intended to include large atmospheric circulations between the North Atlantic, the Mediterranean Sea, and Northern Africa. It also includes important sources of atmospheric emissions located along the English Channel

(United Kingdom, Northern France, Belgium, and the Netherlands) and Northern Africa (Gangoiti et al., 2006a). The second domain (d02), with a resolution of 9 km, incorporates the entire Iberian Peninsula, the South of France, and the coast of Northern Africa. In this way, it can document the Atlantic fronts over the region, the summer anticyclones and associated mesoscale flows in the Western Mediterranean Basin (Gangoiti et al., 2001, 2006b) and the flows developing in the Strait of Gibraltar (in 't Veld et al., 2021;

Massagué et al., 2021). We also included a third domain (d03), with a resolution of 3 km, covering the North of the Iberian Peninsula and the South of France, so that atmospheric flows developed over the Ebro and the Douro Valleys could be represented with an adequate detail. This third domain includes areas of special interest with $O_3$ measuring reference stations for the analysis of interregional $O_3$ transport (Navazo et al., 2008; de Blas et al., 2019; Gómez et al., 2020). We used 31 η layers covering up to approximately



15,500 m AGL. The vertical resolution near the boundary layer, at the surface, is greater than at higher
levels, where the distance between layers increases (Table S1).

**Table 1. Spatial characteristics of the domains used in WRF and CAMx.**

| Domain | Spatial Resolution | WRF Number of cells | CAMx Number of cells |
|--------|--------------------|---------------------|----------------------|
| d01 | 27 km x 27 km | 162 x 162 | 160 x 160 |
| d02 | 9 km x 9 km | 195 x 150 | 193 x 148 |
| d03 | 3 km x 3 km | 393 x 186 | 389 x 182 |

The physical parameterizations of the meteorological model are determinants when simulating air quality.
In this study, we proposed a configuration already used by other studies in IP with satisfactory results
(Borge et al., 2014; Escudero et al., 2019). Other studies, also carried out in IP (Pay et al., 2010; Borrego
et al., 2013; Banks and Baldasano, 2016), have used similar parameterizations with changes in the
configuration of the Planet Boundary Layer (PBL). The selected parameterization is shown in Table S2. It
is mainly based on the configuration of Borge et al. (2008) modifying the longwave radiation scheme by
the Rapid Radiative Transfer Model (RRTM) (Mlawer et al., 1997) recommended by the WRF developers.
Additionally, Sea Surface Temperature (SST) supplied by NOAA has been used, specifically Optimum
Interpolation SST (https://www.ncei.noaa.gov/products/optimum-interpolation-sst), with a spatial
resolution of 0.25° x 0.25° and a daily temporal resolution (Banzon et al., 2016).

Initial and boundary conditions were generated using 6 h reanalysis from the European Centre for Medium-
Range Weather Forecasts (ECMWF), specifically, ERA-Interim reanalysis global data (Berrisford et al.,
2011) of 0.75° x 0.75° horizontal resolution. Its vertical resolution is higher near the surface (every 25 hPa
from 1,000 hPa to 700 hPa), decreasing for higher levels. We also used the same data for Four-Dimensional
Data Assimilation (FDDA) above the PBL in the coarser domain (d01).




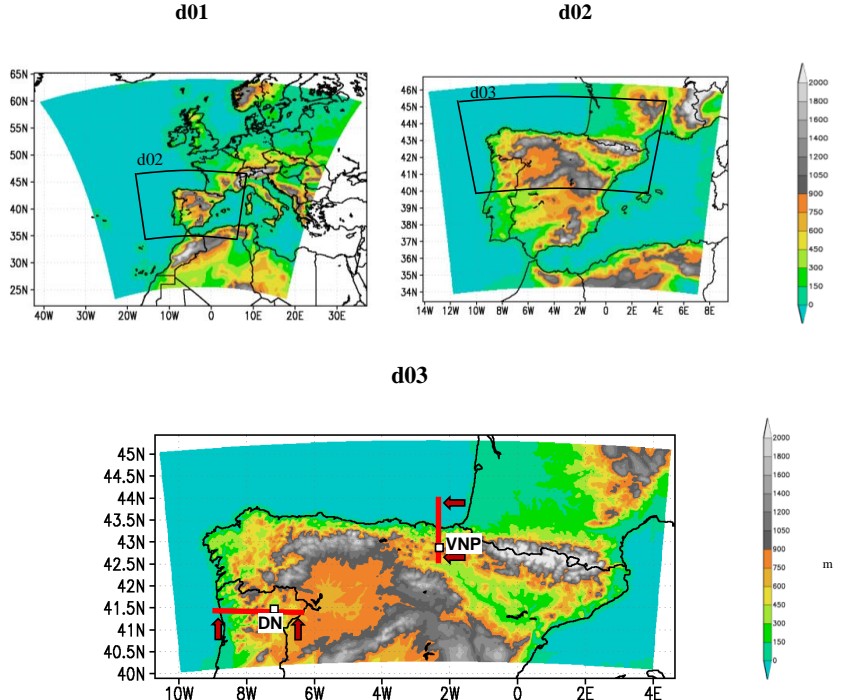

**Figure 2. Domains used for the meteorological and photochemical simulations on its topographic map: d01 (27 km) -Europe and North Africa, d02 (9 km) - Iberian Peninsula-, d03 (3 km) -Northern Iberian Peninsula-. Red lines: location and extent of atmospheric vertical cross-sections analyzed in VNP and DN.**

**2.1.2.    Photochemistry and dispersion**

We used the Comprehensive Air Quality Model with Extensions (CAMx), version 6.50 (Ramboll Environment and Health, 2018). The domain and horizontal resolution selected for this air quality model were identical to those used for WRF model (see Table 1). We used 14 σ layers going up to approximately 4,800 m AGL with the first layer being approximately 20 m thick. Concentrations were calculated at the

midpoint of each layer, so the modeled values of the first layer corresponding to a height of approximately 10 m AGL, and the same applies to other layers. The different thicknesses of the layers and their correspondence with WRF layers are shown in Table S3.

The gas-phase mechanism CB6r4 was used in this work (Ramboll Environment and Health, 2018). For inorganic thermodynamics and gas-aerosol partitioning CAMx uses ISORROPIA (Nenes et al., 1998, 1999)

and for dry deposition we chose the algorithm of Zhang et al. (2001, 2003). The $O_3$ column data was obtained from the $O_3$ Monitoring Instrument (OMI) of NASA's Total $O_3$ Mapping Spectrometer (TOMS) satellite, which has a daily temporal resolution and a horizontal spatial resolution of 1° x 1° (available at https://acd-ext.gsfc.nasa.gov/anonftp/toms/omi/data/ozone/). $O_3$ column data were used in the Tropospheric Ultraviolet and Visible (TUV) radiation and photolysis model used by CAMx: Dr.

Madronich's preprocessor for CAMx calculates the photolysis rates for clear skies, and then CAMx internally adjusts these rates in case of clouds or aerosols (NCAR, 2011). The initial and boundary conditions for the first domain (d01) were obtained from the global air quality model CESM2.1: CAM-Chem (Lamarque et al., 2012). We first ran the first domain (d01), and for the other two (d02 and d03) we





used BNDEXT CAMx preprocessing program to generate initial and boundary conditions extracted from
d01. In the first simulation for d01, we extracted the following simulated pollutants' concentrations to
generate d02's initial and boundary conditions: $O_3$, NO, $NO_2$, $SO_2$, $OH^\bullet$, $HO_2^\bullet$, $H_2O_2$ (hydrogen peroxide),
CO, $CH_4$, Ethane, Ethene, Ethyne, Propane, Formaldehyde, Isoprene, Monoterpenes, Benzene, Toluene and
other monoalkyl aromatics, Xylene and other polyalkyl aromatics, $HNO_3$, HONO (nitrous acid), PAN
(Peroxyacil Nitrate), and $NO_3^\bullet$.

The CAMx domain configuration was the same as that used in WRF (Figure 2) and the type of projection,
their central points, and reference latitudes. However, a slight reduction of dimensions was necessary for
its correct usability in CAMx, due to the way CAMx domains are configured and to the limitation imposed
by some emission models such as MEGAN. Also, for CAMx to properly solve the boundary conditions of
the nested domains, some cells, denominated as buffer cells, were added at the outer edges around the
perimeter of each domain.

CAMx incorporates the WRFCAMX preprocessor, version 4.6, which transforms the WRF meteorological
variable fields into the specific meteorological variables required by CAMx. We chose to run this program
with the YSU scheme of the PBL (Hong, Noh, and Dudhia, 2006) to be consistent with the PBL
configuration used in WRF.

### 2.1.3. Emissions

We used the Model of Emissions of Gases and Aerosols from Nature (MEGAN) (Guenther et al., 2006;
Guenther et al., 2012), an empirical model of biogenic emissions most widely used by the scientific
community for the calculation of VOCs from vegetation (Sindelarova et al., 2014). The new version
(MEGAN 3.0) includes for the first time a processor for calculating Emission Factors (EF) for different
species, where the user may incorporate custom high spatial resolution EF databases from specific
vegetation data. This processor contains a wide selection of EFs for more than 42,500 species types based
on the available databases (Guenther, 2017).

MEGAN requires as input for the biogenic emissions estimation for the different domains used in the
CAMx simulation a meteorological simulation, a Leaf Area Index (LAI) spatial distribution, and EF spatial
distributions. We incorporated meteorological data calculated by WRF into MEGAN 3.0 through the
Meteorology-Chemistry Interface Processor (MCIP) tool (Otte and Pleim, 2010), using the preprocessors
included in this new version of MEGAN to calculate EF, but we improved it by updating Spanish land use
and vegetation maps databases from the National Forest Inventory (Torre-Pascual et al., 2021). Of the
existing global LAI products, we chose the one generated by the MOderate resolution Imaging
Spectroradiometer (MODIS) instrument of NASA's Aqua and Terra satellites (Myneni et al., 2002; Yang
et al., 2006). The wide use of this product is due to its high spatial resolution (1 km x 1 km), temporal
resolution (every 8 days) and its frequent updating (Yuan et al., 2011). However, the instrument shows
uncertainties due to cloudiness and seasonal snow cover, and current MODIS LAI products are spatially
and temporally discontinuous and inconsistent (Zuazo et al., 2023). Thus, we used the 2010 reprocessed
MODIS LAI by Yuan et al. (2011), instead of the one for 2018 because it was not available at the time of
this study.

We used the Emission Database for Global Atmospheric Research (EDGAR) global anthropogenic
emission inventory (Crippa et al., 2018), in its version 4.3.2, published in December 2017. EDGAR contains
anthropogenic emissions from the European and African continents, which fit the extension of the main
domain (d01) with a high spatial (0.1° x 0.1°) and temporal resolution (monthly averages for 2010) for the
whole area. We selected the most relevant compounds for the analysis of tropospheric $O_3$ pollution
episodes, specifically: CO, $NH_3$, Non-Methane Volatile Organic Compounds (NMVOC), NOx, $SO_2$, and
$CH_4$. Emissions due to aviation have been excluded, except for landings and take-offs, as they do not
originate near the surface and can be expected to have little influence on surface and near-surface $O_3$. We
used the SPECIATE tool (EPA, 2016) to speciate NOx and NMVOC.



Emissions in EDGAR's inventory are classified according to their origin following the Convention on Long-Range Transboundary Air Pollution (CLRTAP) - Nomenclature for Reporting (NFR) sectors (Janssens-Maenhout et al., 2019). We performed the daily and hourly temporal distribution of emissions using the temporal distribution coefficients used in the LOTOS-EUROS CTM (Denier van der Gon et al., 2011). LOTOS-EUROS temporal profiles were defined for SNAP (Selected Nomenclature for Sources of Air Pollution) sectors contemplated in the CORINAIR/EMEP methodology (EEA, 2016). Therefore, we regrouped the latter sectors based on the NFR-SNAP mapping table (https://www.ceip.at/fileadmin/inhalte/ceip/00_pdf_other/nfr09_snap_gnfr.pdf). We used the Sparse Matrix Operator Kernel Emission model (SMOKE) (https://www.cmascenter.org/smoke/) for spatial disaggregation and adaptation for the domains, temporal disaggregation, and pollutant speciation.

### 2.2. Validation

To validate the WRF-CAMx simulation, we combined the analysis of the modeling results with the assessment of meteorological reanalysis, in particular ERA5 reanalysis, meteorological observations, and $O_3$ measured concentrations. This allowed us to verify the model's performance with experimental data, not only at the surface level but also at upper levels. For comparison with experimental data, we have preferentially taken the higher spatial resolution outputs of the simulations.

### 2.2.1. ERA5 hourly reanalysis

We have taken as a reference for validating the meteorological simulation the ERA5 reanalysis. ECMWF released a new, improved meteorological reanalysis, namely ERA5 (Copernicus Climate Change Service, 2018; Hersbach et al., 2020), with a higher spatial resolution (0.25° x 0.25°) and higher temporal resolution (hourly) than ERA-Interim. Due to the difficulty in collecting meteorological observation data and the unreliability of some of the data found, we have decided to use ERA5 reanalysis as a main reference, in addition to the selection of 4 meteorological stations that we mention afterward, since it incorporates most of the official meteorological measurements made for this region. We have compared ERA5 surface temperature and winds (surface and 750 hPa) with the WRF simulation. With all this information, we have performed qualitative comparisons for a proper understanding of the episode. In addition, we have examined the visible channel images from the Meteosat satellite, available on NOAA's Global ISCCP B1 Browse System (Knapp, 2008), shown in Figure S1, for evaluating cloudiness and the synoptic evolution during the episode.

### 2.2.2. Surface and upper air meteorological observations

Among the meteorological observational data, we have compared radiosonde data from Lisbon, Santander, and Bordeaux (locations shown in Figure 1) with the WRF simulation, allowing us to evaluate the atmospheric evolution at different altitudes throughout the episode. We gathered radiosonde data from the database of the University of Wyoming (http://weather.uwyo.edu/upperair/bufrraob.shtml), as it has an extensive compilation of all radiosoundings conducted globally.

We compiled surface observations data from two stations of the Basque Meteorological Network (EUSKALMET), one of them located in Bilbao (Deusto) at sea level, and another one in Oiz, at 998 m ASL (Figure 1). For Portugal, we collected two stations' data from the Global Hourly - Integrated Surface Database (ISD) of the NCEI (https://www.ncei.noaa.gov/products/land-based-station/integrated-surface-database), one in Lisbon, at sea level, and Penhas Douradas, at 1,398 m ASL.

### 2.2.3. Surface $O_3$ measurements

To analyze air quality in the IP study area, we used surface observation data of hourly $O_3$ concentrations for Spain and Portugal. The database from Spain is available at MITECO Ministry's website (https://www.miteco.gob.es/es/calidad-y-evaluacion-ambiental/temas/atmosfera-y-calidad-del-aire/calidad-del-aire/evaluacion-datos/datos/Datos_oficiales_2018.aspx) which groups all the data from the air quality networks of the Autonomous Communities, and those for Portugal from its Air Quality Network





(https://qualar.apambiente.pt/) provided by the Agência Portuguesa do Ambiente. For the statistical analysis of the two regions analyzed in this paper, we have selected the stations in NAI from the Spanish database shown in Figure 1, and for WAI all the stations from the Portuguese database.

**2.2.4.    Statistical evaluation of simulated O₃ concentrations**

The uncertainty associated with the models is determined by comparing the experimental data (measurements) and the results of their simulations. Several studies have developed different methodologies and there is currently no standardized methodology for this purpose (Borrego et al., 2008). In recent years, the use of a series of statistical indicators has prevailed in the scientific literature (Bessagnet
et al., 2016; Oikonomakis et al., 2018). In this work, we have chosen to use a set of metrics commonly employed by the aforementioned studies described in Table 2. This has allowed us to compare the metrics used here with the work of other authors. We carried out this evaluation for the selected stations shown in Figure 1 for NAI and WAI, also shown with their coordinates in Tables S4 and S5.

**Table 2. Statistical metrics used for the photochemical simulation.**

| Statistical metrics | Equation |
|---|---|
| Mean Bias (MB) | $$\frac{1}{N}\sum_{i=1}^{N}(Model_i - Obs_i)$$ |
| Mean Error (ME) | $$\frac{1}{N}\sum_{i=1}^{N}|Model_i - Obs_i|$$ |
| Normalized Mean Bias (NMB) | $$\frac{\sum_{i=1}^{N}(Model_i - Obs_i)}{\sum_{i=1}^{N} Obs_i}$$ |
| Root Mean Square Error (RMSE) | $$\sqrt{\frac{1}{N}\sum_{i=1}^{N}(Model_i - Obs_i)^2}$$ |
| Index of Agreement (IOA) | $$1 - \frac{N \cdot RMSE^2}{\sum_{i=1}^{N}(|Model_i - \overline{Obs}| + |Obs_i - \overline{Obs}|)^2}$$ |
| Pearson correlation coefficient (r) | $$\frac{\sum_{i=1}^{N}(Model_i - \overline{Model}) \cdot (Obs_i - \overline{Obs})}{\sqrt{\sum_{i=1}^{N}(Model_i - \overline{Model})^2} \cdot \sqrt{\sum_{i=1}^{N}(Obs_i - \overline{Obs})^2}}$$ |




## 3. Results and discussion

First, we have evaluated the results of the meteorological model as they determine the performance of the photochemical model. Secondly, we have analyzed the evolution of the episode from CAMx results and
contrasted it with observations of O₃ concentrations. At the end of this section, we have also included a statistical analysis of the concentrations to assess the performance of CAMx.

### 3.1. Meteorology

The six-hourly NCEP Climate Forecast System Reanalysis (CFSR) historical archive in Wetterzentrale (http://www.wetterzentrale.de/) and the ERA5 reanalysis are used to describe the synoptic meteorology.
Surface and upper air meteorological observations of a set of stations (section 2.2) are also discussed in this section in the context of the different scales of the meteorological processes working together during the episode and the eventual adequacy of the response of the WRF model to the observed meteorology.

### 3.1.1. Synoptic analysis and upper-level winds

Following the NOAA NCEP Climate Forecast System Reanalysis (CFSR) (Saha et al., 2014), the synoptic
conditions during the O₃ episode (2-6 August) (Figure 3) were characterized by a large upper-level ridge which extended from northern Africa to western Europe crossing the Iberian Peninsula and an associated large area of surface high-pressures. The surface anticyclone covered the whole European Atlantic coast from Scandinavia to Iberia. This pressure distribution is compatible with East to Northeasterly winds at surface levels following the coast along the European Atlantic, and warm southerlies at upper levels over
IP, which bring vertical stability and the adequate conditions to O₃ episodes ("accumulation periods") in central Iberia (Querol et al., 2018). The easterly winds in the marine boundary layer of the northern coast of Spain and the sea-breeze inland convergences are behind most of the O₃ episodes in the Basque Country (Gangoiti et al., 2006a). These episodes, though less intense, occurred even during the COVID-19 lockdown period after a significant reduction of anthropogenic precursors (Gangoiti et al., 2021), and they
were attributed to O₃ importation across southern France and the Bay of Biscay.

The ERA5 reanalysis at the surface and upper levels shows a more detailed wind field distribution and time evolution during the initiation of the episode (Figure 4). Easterly winds in the Bay of Biscay and the northerlies at the coast of Portugal (Figure 4 a.1 and b.1) in the marine boundary layer were decoupled from the relatively warm winds at upper levels (Figure 4 a.2 and b.2). That air mass circulated anticyclonically
around IP, a fact that could hardly be inferred from the CFS Reanalysis in Figure 3, forcing moderate southerlies over WAI, weak westerlies over NAI, and almost calm conditions over the SW coast of France on August 2 (Figure 4 a.2). That meant a maximum wind directional shear of 180º in both WAI and NAI regions. This type of upper-air anticyclonic circulations seem to be a key component of the Northern African middle troposphere wind regime, which is behind the desert dust transport accumulation and
redistribution in the region (Gangoiti et al., 2006b). From August 3 onwards, there was a change in the wind field at upper levels, registered by the ERA5 reanalysis: the warm air mass circulating at upper levels moved to the west, and the wind turned to the N over the eastern half of IP. That changed completely the atmospheric circulation at those levels, being opened to the entry of air masses of European origin to NAI, while the circulation pattern remained from the south over a larger part of the coastal WAI and from the
NE over Southern Portugal (Figure 4 b.2). However, these changes were not observed at surface level: at the Atlantic coast of Iberia surface winds did not show significant changes from the previous day (Figure 4 b.1). The new wind configuration at surface and upper levels over WAI, which lasted for the rest of the O₃ episode, was more similar to those described in Gangoiti et al. (2006a) and Valdenebro et al. (2011) for the northern coast.

The end of the episode started on August 6 with the development of an upper-level trough associated with a mid-latitude depression to the south of Iceland (Figure 3). The trough, marked with a black arrow in the figure, extended to NW Iberia and forced a surface (and upper level) cold air mass advection from the north-west with a frontal region (dashed line in Figure 3), which crossed NAI and WAI during the following





24 hours. The ERA5 wind and temperature data in Figure 5 shows the observed changes during the frontal
passage: the west and northwesterly wind advection started on August 6 and moved to the west during the
following day, with westerlies at the surface and intense south-westerlies at upper levels (Figure 5). A
similar wind field distribution has been estimated by the WRF simulation during the whole episode, both
at surface and upper levels, shown in Section 3.2. in the context of the simulated inter-regional O₃ transport
and distribution.

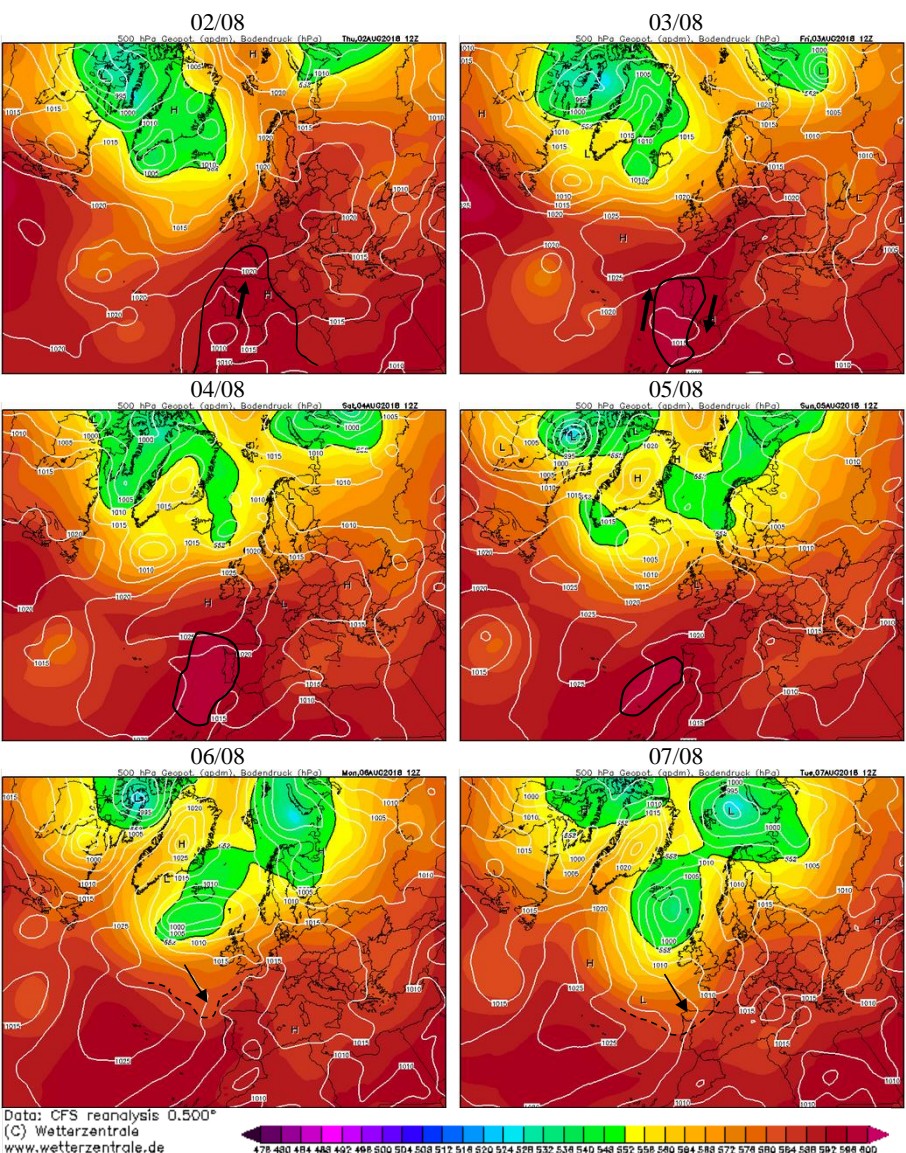

**Figure 3. Synoptic conditions during the O₃ episode with geopotential height at 500 hPa (geopotential dm,
shaded) and surface pressure (hPa, contours) during the period of the high ozone episode (August 1-7). L and H
mean "low pressure centre" and "high pressure centre" respectively. Continuous black lines represent the
warmer air mass over the IP and dashed black lines represent the Atlantic advection. Source: NCEP CFS
reanalysis from www.wetterzentrale.de.**



Radiosonde wind data and WRF simulated vertical profiles at three sounding sites (Lisbon, Santander, and Bordeaux) of WAI and NAI are represented in Figure 6 for the "extended" $O_3$ episode (1-8 August). Observations (left) and modeled vertical winds (right) agree and correspond with the surface and upper air wind field reanalysis described above. During the initiation of the episode (1-2 August) southerly winds (SW at Santander and Bordeaux, SE in Lisbon) blew above 1,500-2,000 m ASL, decoupled from the

easterlies at the surface (below 1,000-1,500 m ASL). The following changes in the wind field at upper levels, registered by the reanalysis during the period 3-6 August, are represented by the northerlies above 1500-2000 m ASL (Santander and Bordeaux) and the easterlies backing to the north (Lisbon), depicted in Figure 6. These changes correspond to the westward displacement of the anticyclonic circulation at upper levels. The cold front passage at the end of the episode is represented in Figure 6 by the surface north-

westerlies backing to the SW with height, initiated during the afternoon of August 6 in Lisbon and Santander, and on August 7 in Bordeaux.

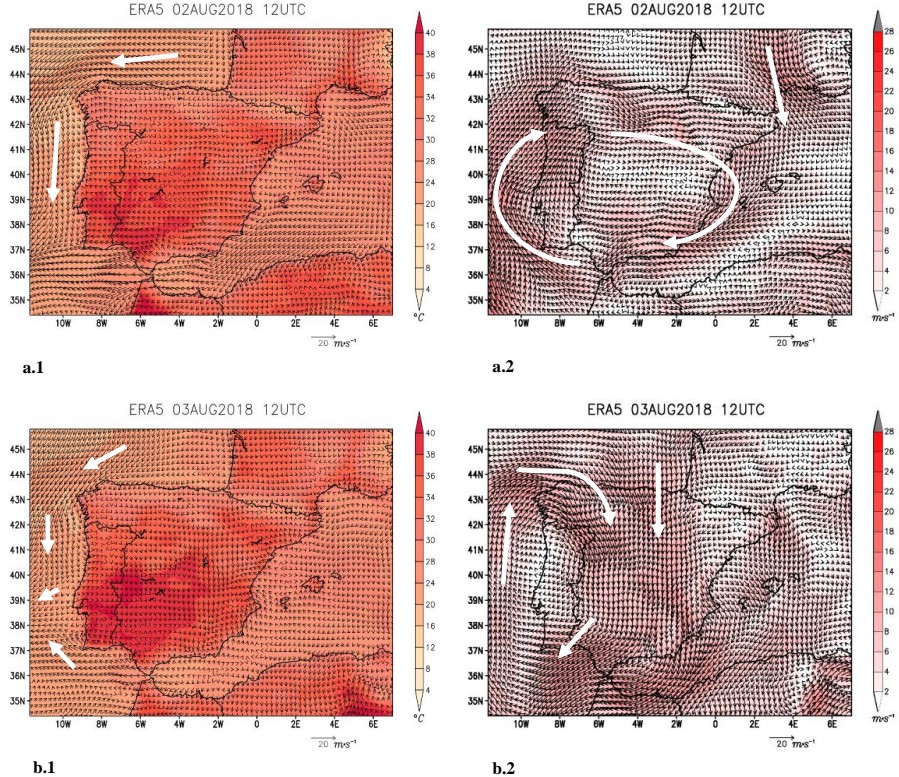

**Figure 4. ERA5 reanalysis for August 2 and August 3. Left panels: Surface wind field vectors and surface air temperature (color scale). Right panels: wind field vectors at 750 hPa and wind speed at 750 hPa (color scale).**



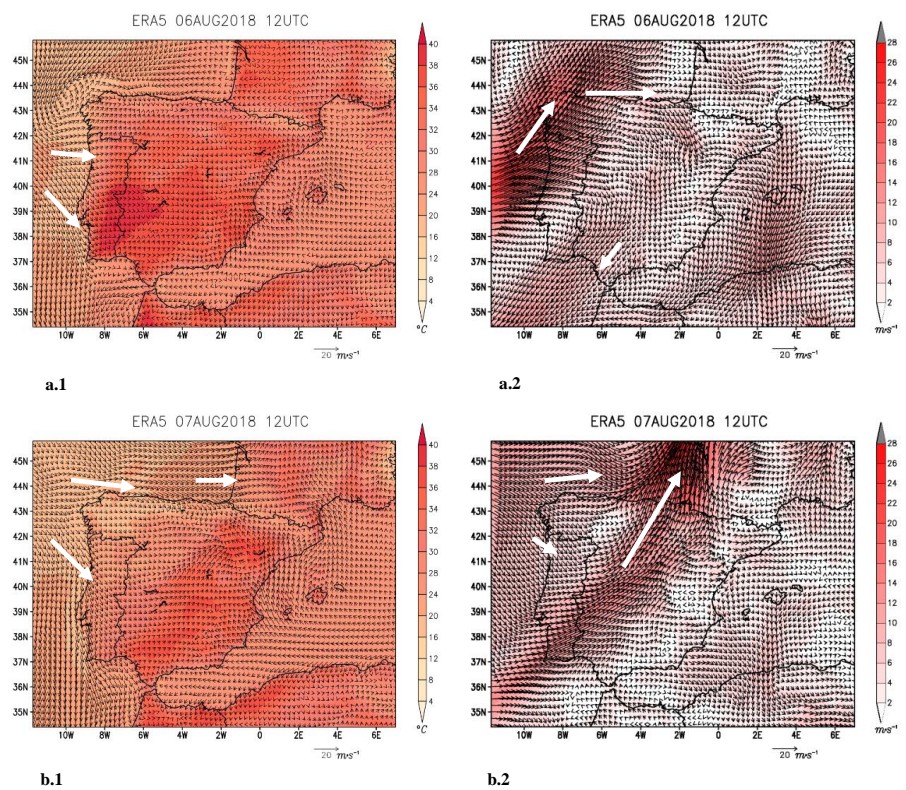

**Figure 5. ERA5 reanalysis for August 6 and August 7. Left panels: Surface wind field vectors and surface air temperature (color scale). Right panels: wind field vectors at 750 hPa and wind speed at 750 hPa (color scale).**





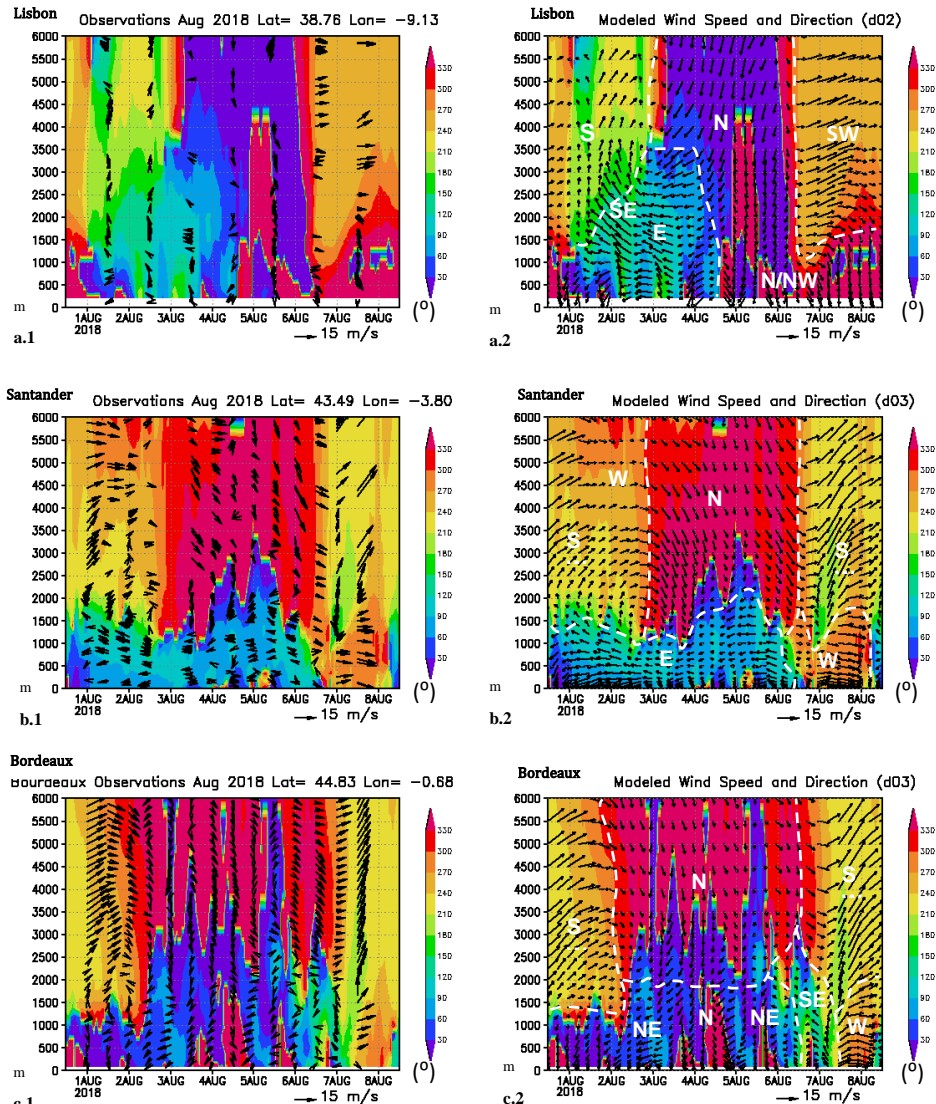

**Figure 6. Wind vectors measured in radiosonde (left) and WRF simulations (right) for the period 01-08 August 2018. The range of colors in all graphics represents the simulated wind direction. The size of the vectors represents wind speed.**

### 3.1.2. Surface temperature and winds

Figures 7 and 8 show the sequence of temperatures and wind observations (red) and WRF simulations (blue) at the two selected surface meteorological stations in NAI during the extended $O_3$ episode (1-8 August). They document the meteorological conditions at a sea-level coastal station (Deusto) and at an elevated inland site (mount Oiz). Deusto is located in the city of Bilbao, in a coastal valley with SE-NW orientation, draining directly into the sea along a 10-km-long estuary. Thus, the land-sea breeze regime at this station is represented with successive channeled land (S and SE) and sea (N-NW) daily wind cycles.



The coastal convergence contrasts with the meteorological conditions at the inland station (Figure 8), which
was not affected by sea breeze regimes. The wind sequence in mount Oiz is more similar to the upper air
observations at around 1,000 m ASL over the Santander sounding site (Figure 6), located 100 km to the
west in the northern coast. The simulations follow main temperature and wind shifts during the episode in
both stations. Two main changes can be distinguished in Figure 8. (1) During the first day of the episode,
the south-easterly winds changed to the north-east, concurrent with the observed changes in the upper-level
anticyclonic circulation described above and persisted during the rest of the $O_3$ episode. The simultaneous
documented convergence of the coastal sea-breeze regimes shown in Figure 7 (transporting local emissions)
together with the E-W transport, in the marine boundary layer of $O_3$ and precursors originated further away
to the East was responsible for the observed $O_3$ concentrations in the inland monitors during that period, as
discussed in the next section. (2) During the last day of the episode (August 6), intense south-westerly
prefrontals preceded the arrival of the cold front (NW in Figure 8) at the end of the day in mount Oiz. Those
warm (30 ºC) offshore prefrontals, registered at mount Oiz at around 1,000 m ASL, rose the temperature at
the coastal stations (37 ºC in Deusto at midday), when the upper-level southerlies were coupled with the
surface winds at the lee of the coastal mountain ranges, as it was the case during that foehn episode in the
Basque Coast. Attending to the coastal station records (Figure 7), the sea breeze could develop against the
offshore winds during the afternoon, while the prefrontal south-westerlies still kept blowing above the
coastal sea breeze and on top of the inland mountain stations, as mount Oiz (Figure 8).

Figures 9 and 10 show a similar sequence (as in Figures 7 and 8) of temperatures and wind observations-
simulations in two meteorological surface stations in WAI. Similar to the NAI site selection, they document
the meteorological conditions at a sea-level coastal station (Lisbon) and at an elevated inland site (Penhas
Douradas, 1,398 m ASL). The simulations also follow the main temperature and wind shifts during the
period in both stations. The land-sea breeze regime in Lisbon was represented with successive land (E and
NE) and sea (NW) daily wind cycles. As for the case of the NAI stations, the sites showed a completely
different behavior, mainly due to the observed decoupling between the upper and lower-level flows. Sea
breeze cycles were observed at the coastal station in Figure 9, which persist during the whole episode. On
the contrary, initial south-easterly winds in Penhas Douradas (Figure 10), in agreement with the upper-level
anticyclonic circulation, changed to the north and north-west, according to the observed synoptic changes
(Figure 4) and the vertical soundings in Lisbon (Figure 6) between 1,000-1,500 m ASL. During the last day
of the episode (August 6), a temperature decrease of 10-15 ºC and intense and persistent northwesterly
winds (without cycles) in both stations (Figures 9 and 10) marked the cold front advection before midday,
preceding the changes observed in the NAI stations.



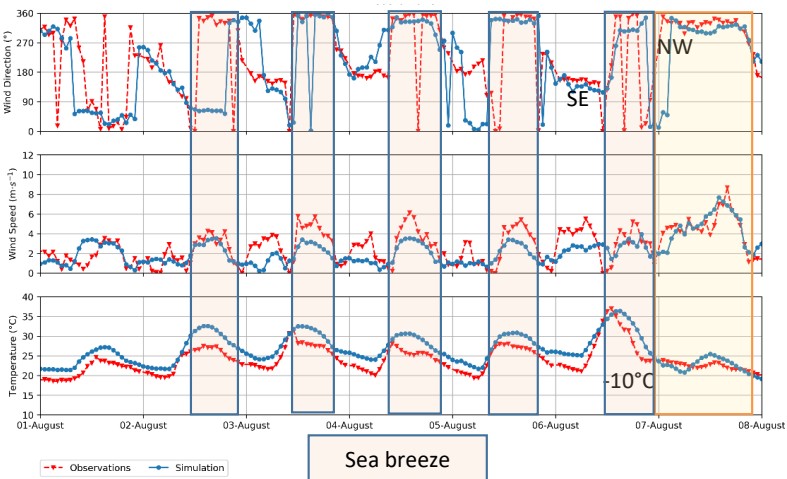

**Figure 7. Comparison of observed (red) and simulated (blue) wind direction, wind speed, and temperature time series for the Deusto station (August 1-7, 2018). The phases of sea breezes are marked in blue squares and the passage of the front with its consequent reduction of temperatures and wind shift to northwest in orange.**

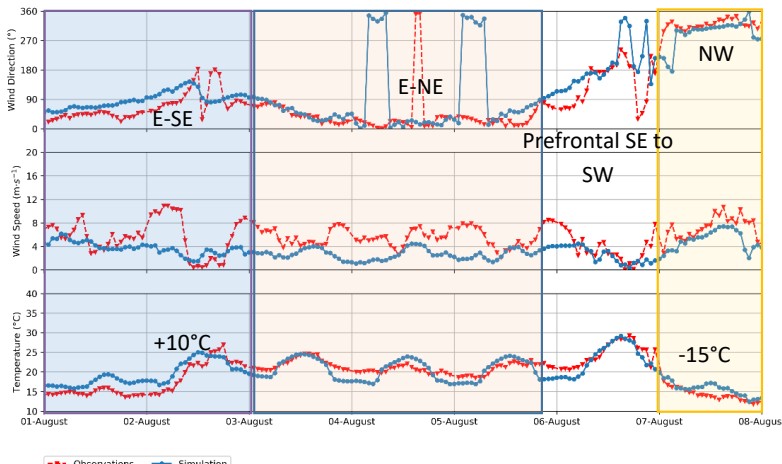

**Figure 8. Comparison of observed (red) and simulated (blue) wind direction, wind speed, and temperature time series for the Oiz station (998 m ASL) (August 1-7, 2018). The three meteorological changes observed during the episode are distinguished by color.**




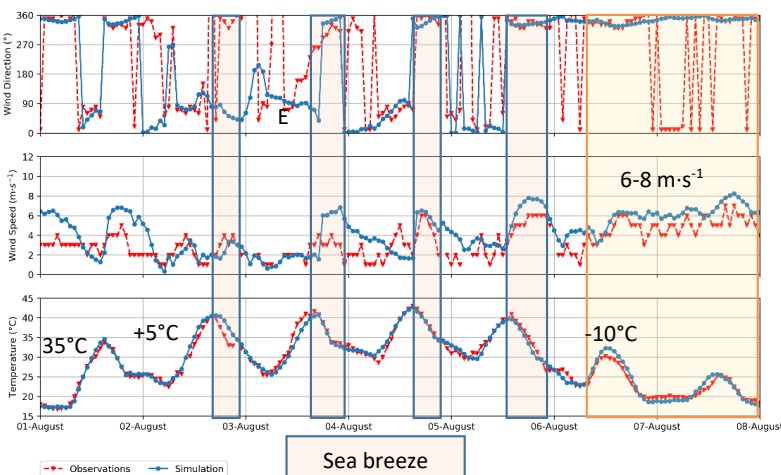

**Figure 9. Comparison of observed (red) and simulated (blue) wind direction, wind speed, and temperature time series for the Lisbon station (August 1-7, 2018). The phases of sea breezes are marked in blue squares and the passage of the front with its consequent reduction of temperatures and wind intensity increase in orange.**

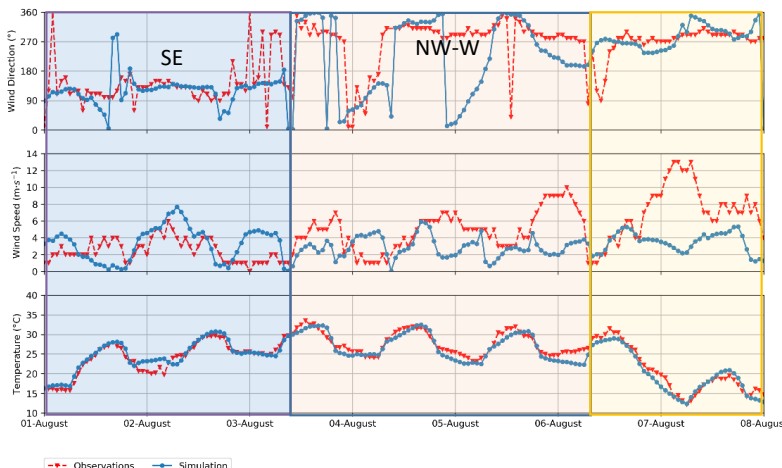


**Figure 10. Comparison of observed (red) and simulated (blue) wind direction, wind speed and temperature time series for the Penhas Douradas station (1398 m ASL) (August 1-7, 2018). The three meteorological changes observed during the episode are distinguished by color.**

**3.2.    O₃ concentrations**

This section presents an analysis of the observed and simulated $O_3$ surface concentrations and their integrated concentrations up to 2,500 m AGL. We have also analyzed two vertical cross-sections (red lines in Figure 2) of the atmosphere in the areas of interest of the Valderejo Natural Park (VNP) and Douro Norte (DN), which have shown some of the main $O_3$ transport pathways in these inland areas where large

exceedances occur.



The evolution of the episode is shown in Figure 11 and Table 3. From July 31 to August 8, O₃ concentrations exceedances in the Iberian Peninsula were numerous (Figure 11), with high concentrations every day in Madrid and Barcelona metropolitan areas. The days with the highest number of measurement stations exceeding the European Directive target value and the information threshold occurred from August 2 to 6,
particularly in the Atlantic Coast of IP (Table 3), where there was a notable increase in concentrations (Figure 11).

**Table 3. Number of air quality monitoring stations within Portugal and Spain and the Atlantic Coast in which the European Directive O₃ target value and information threshold are exceeded, between July 31 and August 08, 2018.**

| Air quality monitoring stations: Portugal and Spain | | | | | | | | | |
|---|---|---|---|---|---|---|---|---|---|
| | 31.07 | 01.08 | 02.08 | 03.08 | 04.08 | 05.08 | 06.08 | 07.08 | 08.08 |
| Number of stations where max 8h-mean concentrations > 120 μg·m⁻³ | 53 | 79 | 125 | 146 | 164 | 203 | 170 | 75 | 31 |
| Number of stations where max 1h-mean concentrations > 180 μg·m⁻³ | 2 | 12 | 7 | 14 | 20 | 9 | 3 | 1 | 0 |
| Air quality monitoring stations: Atlantic Coast | | | | | | | | | |
| Number of stations where max 8h-mean concentrations > 120 μg·m⁻³ | 0 | 1 | 27 | 38 | 31 | 23 | 16 | 3 | 0 |
| Number of stations where max 1h-mean concentrations > 180 μg·m⁻³ | 0 | 1 | 4 | 11 | 13 | 2 | 0 | 1 | 0 |


The initiation phase of the episode on the Atlantic Coast of IP began on August 2. It was characterized by an O₃ maximum hourly concentration increase of more than 40 μg·m⁻³ in Portugal (see Chamusca and Noia stations in Figure 12) and more than 30 μg·m⁻³ in non-coastal areas of NAI (see Valderejo and Los Tojos stations in Figure 12). That increase was due to a fumigation described in Section 3.2.1. On August 3 the
highest number of exceedances of the European Directive occurred in NAI, marking the beginning of the peak phase: 38 stations exceeded the target value and 11 exceeded the information threshold (Table 3). During that second day, O₃ concentrations increased again by more than 30 μg·m⁻³ (Figure 12), with notable increases in DN and VNP of more than 60 μg·m⁻³, reaching maximum hourly concentrations of more than 180 μg·m⁻³. These increases coincided with the beginning of inflows of European continental air masses
into IP with northerly winds. Hourly concentrations above 120 μg·m⁻³ were exceeded daily during this peak phase, extended until August 5.

On August 6, the dissipation phase began in WAI, but not in NAI, particularly in its coastal areas due to the foehn effect described in Section 3.1.2. Finally, on August 7, all concentrations dropped significantly due to the Atlantic advection. The detailed analysis of the phases with the simulated O₃ concentrations is
shown below.



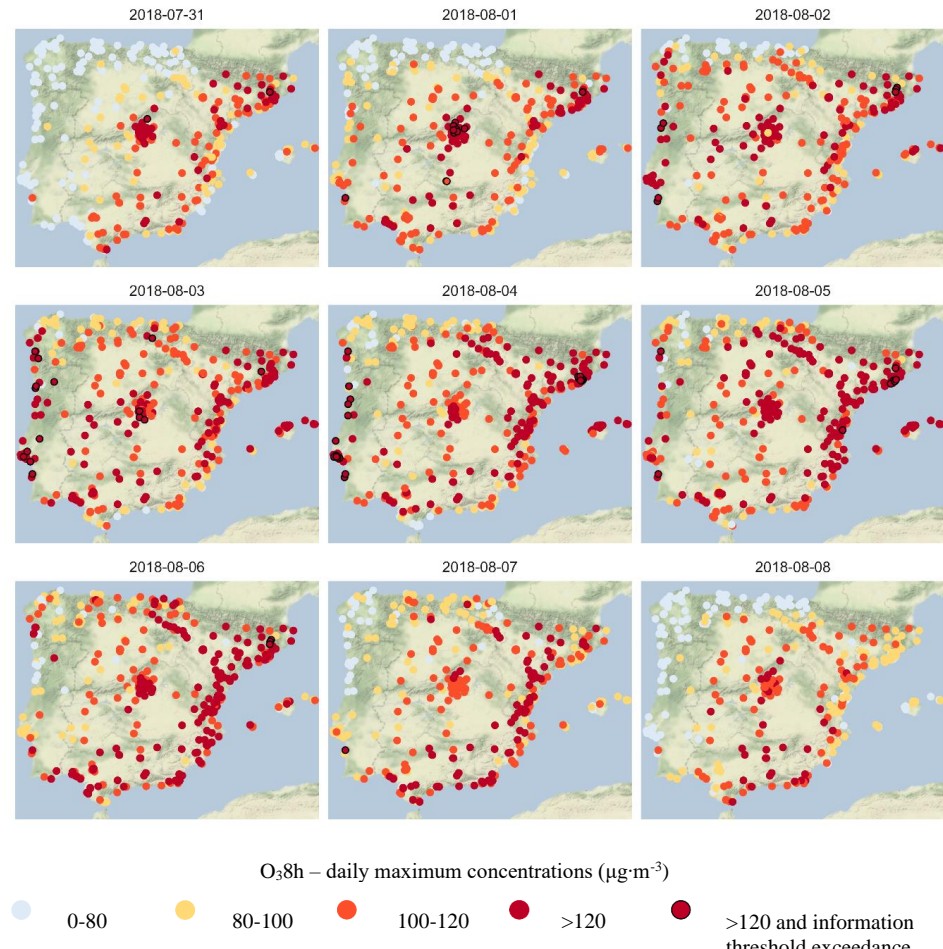

**Figure 11. Daily evolution of the spatial distribution of maximum daily 8-hour O₃ concentrations, from July 31 to August 8.**




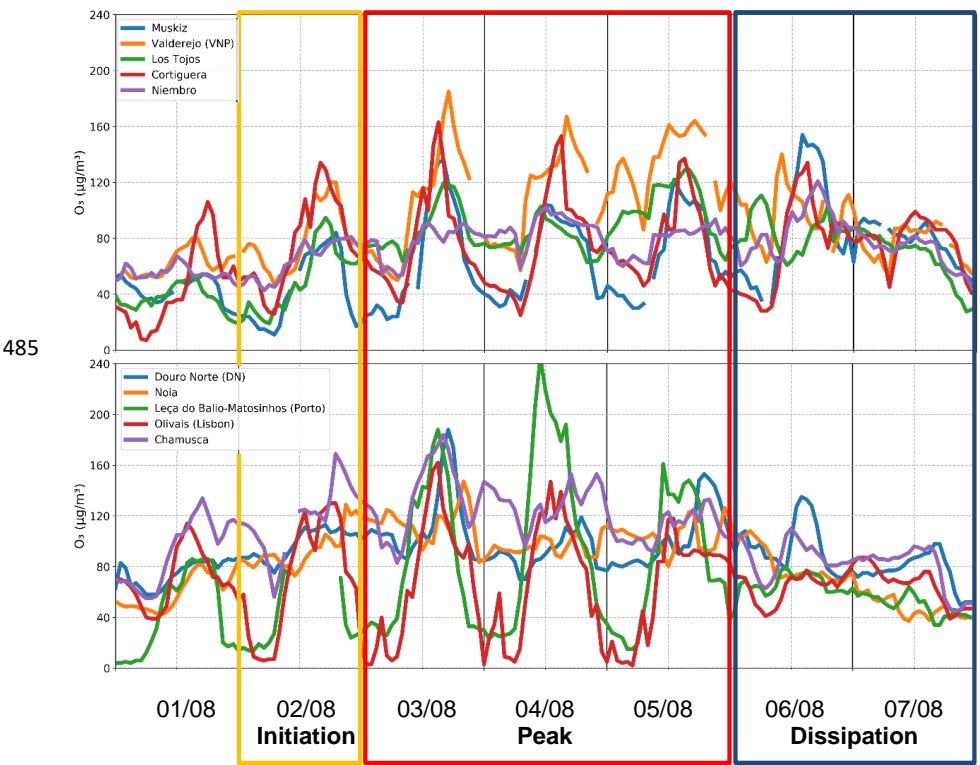


**Figure 12. Ozone hourly concentrations time sequences for 1-7 August 2018 at a selection of stations along Northern Atlantic Iberia (top) and Western Atlantic Iberia (bottom).**


### 3.2.1. Initiation

During August 1, an accumulation of O₃ integrated up to 2,500 m AGL of more than 135 ppm·m in the center of IP, northern coast of Portugal, Western Mediterranean Basin, and NE of IP has been simulated (Figure 13). The winds at altitude, from E and SE, suggest the beginning of the transport of O₃ and other 495 pollutants from E to W of IP. At the surface, the highest concentrations were found in the simulation towards NW of Madrid and N of Barcelona due to the impact of emissions from these metropolitan areas, a pattern that is constantly repeated throughout the episode. Concentrations in WAI began to rise, up to 95-105 µg·m⁻³, while in NAI remained low with 75-85 µg·m⁻³, probably due to a lower photochemical production under cloudier skies (see Figure S1).

On August 2, the air recirculation in the upper-layers, with completely clear skies and stagnant winds, caused an increase in surface O₃ concentrations exceeding 130 µg·m⁻³, compared to the approximately 100 µg·m⁻³ from the previous day (Figure 13). That increase of more than 30 µg·m⁻³ over surface simulated concentrations, and more than 40 µg·m⁻³ over measured maximum concentrations both in NAI and WAI (Figure 12), together with the displacement of the high-altitude O₃-rich air masses towards NAI and WAI, 505 support the hypothesis of fumigation of pollutants as the main cause of the observed surface ozone increases during that day. We have analyzed vertical atmosphere cross-sections in VNP and DN to address this hypothesis.



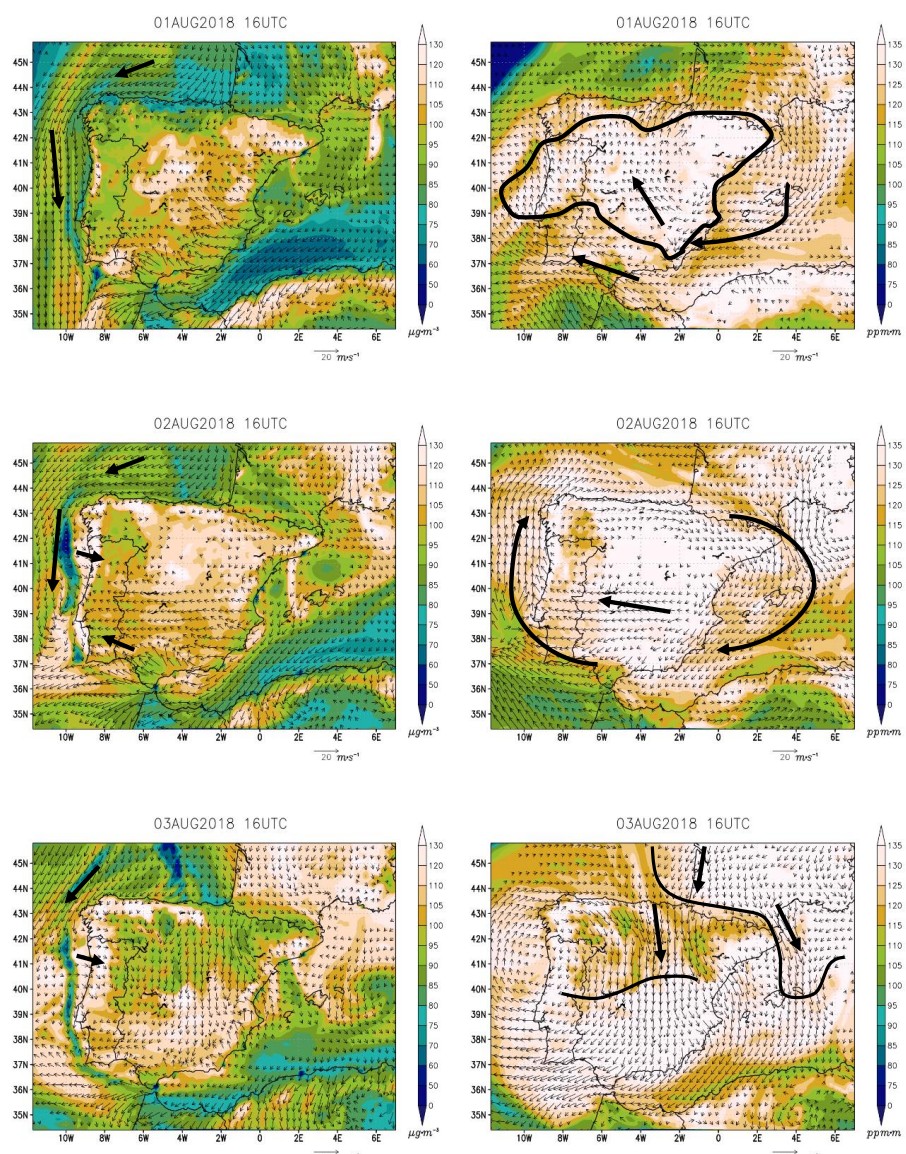

**Figure 13. Simulated O₃ concentrations (color scale) and wind fields (vectors) by WRF-CAMx in d02 at 16 UTC on August 1, 2 and 3, 2018. Left panels show the ozone and wind concentration in surface and right panels the integrated ozone concentration up to 2500 m AGL and wind at 1250 m AGL. Winds lower than 2 m·s⁻¹ have been omitted.**

In VNP, on August 1, the upper polluted air mass was located above 1,800 m ASL according to the simulated O₃ concentrations (Figure 14). On August 2, from 12 UTC onwards, mixing of that high-altitude polluted air mass with the surface occurred in the inland valleys, with simulated O₃ concentrations above 110 µg·m⁻³ (55 ppb in Figure 14). Above 500 m ASL horizontally projected winds were from the S, while below 500 m ASL sea breezes prevailed near the coast, and O₃ concentrations were not significantly high




in coastal areas (80 µg·m⁻³). In the afternoon of August 2, horizontally projected winds above 1,000 m ASL turned to the N and began withdrawing the polluted air mass towards the S of IP. Therefore, the first peak recorded in Valderejo on day 2 (Figure 12) was due to that fumigation process.

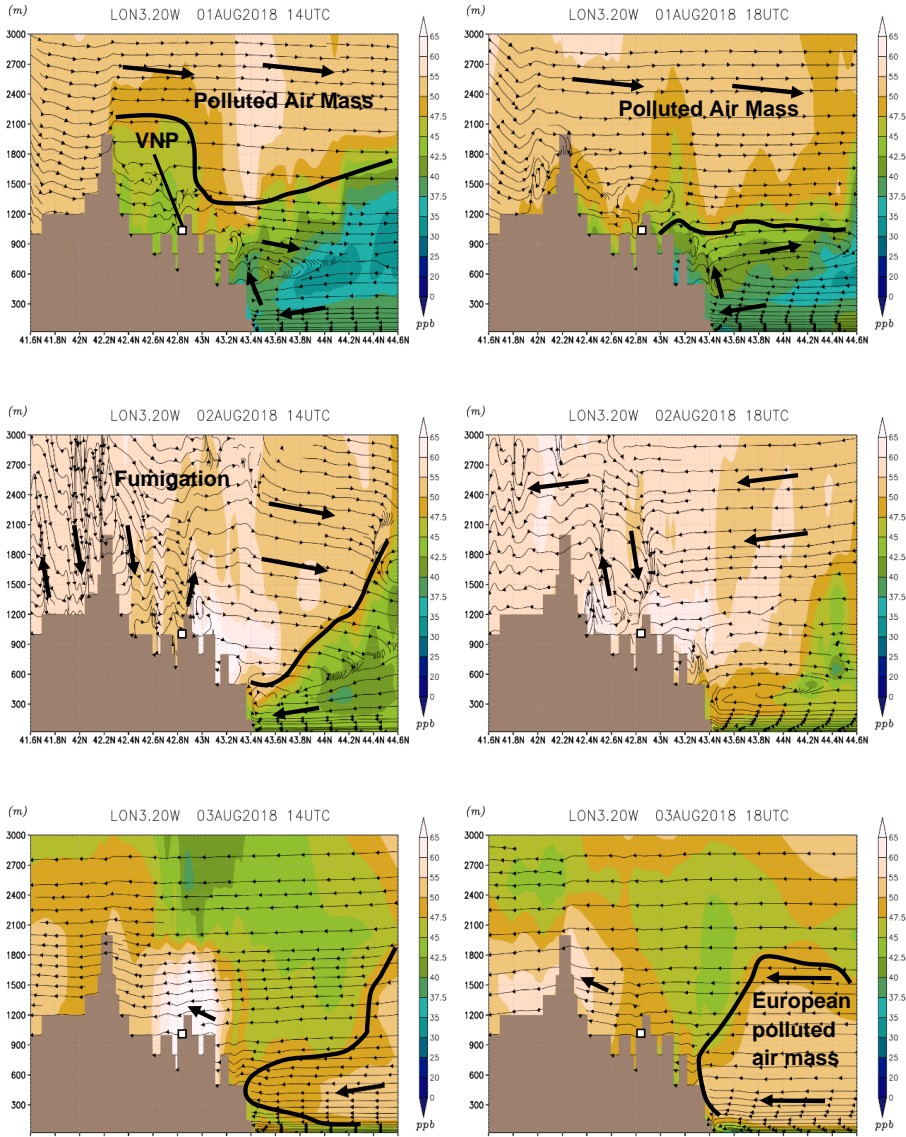

**Figure 14. Simulated vertical O₃ concentrations (color scale) and projected wind fields (stream composed by projected u with w(*10)) by WRF-CAMx in d03 at 14 UTC and 18 UTC on August 1, 2 and 3, 2018, for the VNP vertical cross-section. Concentrations are depicted in ppb as they are altitude independent (1 ppb ≈ 2 µg·m⁻³ at sea level).**





In DN, the upper-level polluted air mass entered at approximately 1,800 m ASL during August 1, with S-SE origin (Figure 15). Low $O_3$ concentrations were observed over the sea surface, while on the coastline there was a slight increase in concentrations in the lower layers between 400 and 1,000 m ASL due to the transport of locally emitted pollutants towards the inland areas pushed by sea breezes with northwesterly
winds. In the inland valleys, high $O_3$ concentrations were registered due to the fumigation of the polluted air masses at altitude, due to convective movements during the afternoon (Figure 15). The coastal return flows during the afternoon, at 900 m ASL (Figure 15), could transport part of that fumigated $O_3$ to the sea surface, generating an $O_3$ reservoir for the following days, similar to the processes described for Mediterranean areas (Millán et al., 2002).

In addition to the fumigation process, surface winds during midday on August 1 and 2 were northerly over WAI and northeasterly over the Bay of Biscay. That could infer a transport of pollutants from the French Atlantic coast towards WAI in the lower layers of the atmosphere (Figure 13) as documented in Gangoiti et al. (2006a), although that transport was more evident from August 3 onwards.

Those two possible $O_3$ transport pathways (fumigation and regional transport) would be responsible for the
significant increase in $O_3$ concentrations in WAI, up to 160 µg·m$^{-3}$ measured inland during the afternoon on August 2 (see Chamusca station in Figure 12) with the onset of the sea breezes (Figure 13). The plume generated on the coast was transported inland and injected through orographic chimneys to the existing recirculating air mass in upper layers, reaching up to 2,400 m ASL.

### 3.2.2.    Peak

During August 3, the change in synoptic conditions led to the transport of the polluted air mass from N to S of the peninsula (Figure 13). At the same time, the SE winds from the previous day over the SW of IP dragged part of the polluted air mass toward the coast of Portugal, causing an accumulation of $O_3$ over WAI (Figure 13). On the surface, as on August 2, WRF-CAMx-simulated $O_3$ concentrations were above 130
µg·m$^{-3}$ over NAI and WAI (Figure 13). The wind shift to N-NE at the end of August 3 caused the entry of new polluted air masses from France, both through the Bay of Biscay towards NAI and through the Gulf of Lion towards the Mediterranean Sea. That transport pathway corroborates one of the accumulation phase transport pathways proposed by Gangoiti et al. (2006a) and Valdenebro et al. (2011). In this episode, however, there was no previous gradual accumulation, it already started with an abrupt rise in $O_3$
concentrations on the previous two days, during the initiation.

Particularly in WAI, during August 2 and 3 there is a notable increase in the $O_3$ simulated concentrations over the sea (Figure 15). We have observed that during August 3 $PM_{10}$ measured concentrations in Western IP increased notably, up to 60 µg·m$^{-3}$. These high concentrations lasted until August 5 (not shown in this paper), and they were concurrent with lower $O_3$ concentrations in upper layers (Figure 15), indicating a
transport of mineral dust from the Sahara Desert to the WAI. Fumigation processes on August 2 would have transported $O_3$ from the upper layers to the surface, whereas on August 3 it would be dust instead of $O_3$ (see the dust location in the satellite map in Figure S1). Those fumigation processes could introduce $O_3$ and PM into the sea-land recirculation cells causing high measured concentrations of both pollutants simultaneously.

In the VNP, over NAI, an intrusion of polluted air through the Bay of Biscay from the North, of French origin, was observed, causing $O_3$ concentrations of 100 µg·m$^{-3}$ (50 ppb) from surface level to 1,500 m ASL (Figure 15). Additionally, we observed how local pollutants emitted on the coastline impacted the inland valleys, producing more than 160 µg·m$^{-3}$ of $O_3$ (Figure S2). Thus, a significant local contribution was added to the already existing regional transport of polluted air masses. In DN, over WAI, the pattern of inflow
with the sea breeze and impact on inland areas was repeated as shown in Figure S3, also documented in other studies (Evtyugina et al., 2007; Monteiro et al., 2012, 2016; Torre-Pascual et al., 2023).



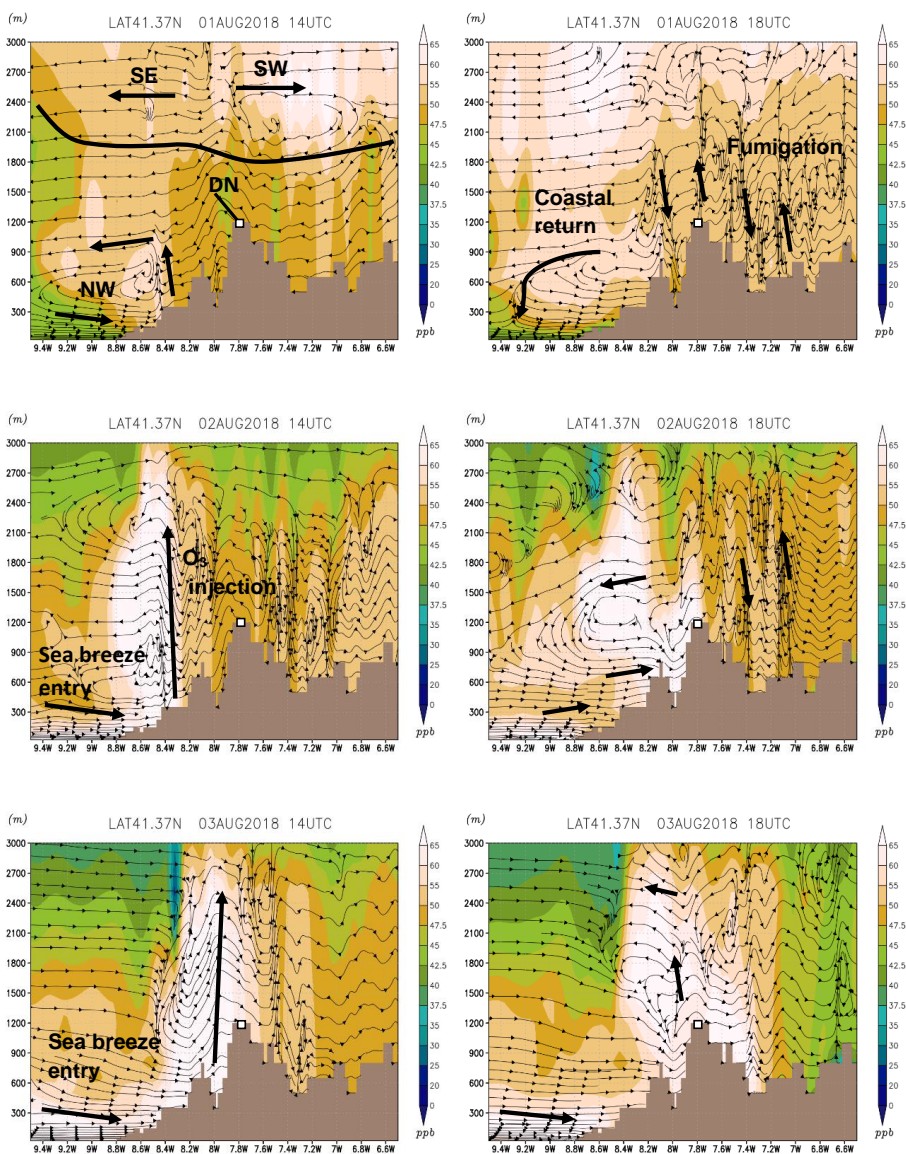

**Figure 15. Simulated vertical O₃ concentrations (color scale) and projected wind fields (stream composed by projected v with w(*10)) by WRF-CAMx in d03 at 14 UTC and 18 UTC on August 1, 2 and 3, 2018 for the DN vertical cross-section. Concentrations are depicted in ppb as they are altitude independent (1 ppb ≈ 2 µg·m⁻³ at sea level).**

During August 4 and 5 the simulated surface O₃ concentrations exceeded again 130 µg·m⁻³ on NAI and WAI (not shown). The transport of pollutants from France to NAI through the Gulf of Biscay and to the Mediterranean Sea through the Gulf of Lion continued during those days and cycles of sea-breezes were repeated. In the case of the Douro Norte station, we observed the transport of polluted air masses from the coast into that area due to the sea-breeze. A sudden rise in O₃ hourly concentrations occurring on August



3, 5, and 6 was caused by the impacts of these air masses, but not with such intensity on August 4 (Figure S3) because of a more Southerly trajectory during that day of the polluted air mass (not shown). The approach of the cold front during August 5 caused prefrontal winds of W-SW component over IP that initiated the transport of all pollutants from the W to the E of IP.

### 3.2.3. Dissipation

The Atlantic air mass entered IP on August 6, introducing cloudiness (Figure S1) and producing wind shift to W over WAI. That change introduced cleaner air to the west of the peninsula (Figure 16). However, in NAI, $O_3$-polluted air masses coming from the W and SW of Iberia were transported with the prefrontal winds, adding $O_3$ to the one previously accumulated days before. That situation caused high simulated $O_3$ surface concentrations ($> 130 \ \mu g \cdot m^{-3}$) and the highest observations over NAI (Figure 11 and Figure 12). The passage of the frontal system generated a simulated "ozone front" confirmed by the measurements. The final entry of cleaner air from the Atlantic Ocean during August 7 significantly reduced $O_3$ concentrations both at the surface and in altitude in the NW region of IP (Figure 16). In the E of IP and the Western Mediterranean Basin, higher concentrations were still found in the simulations, indicating a possible episode during the following days in that territory.

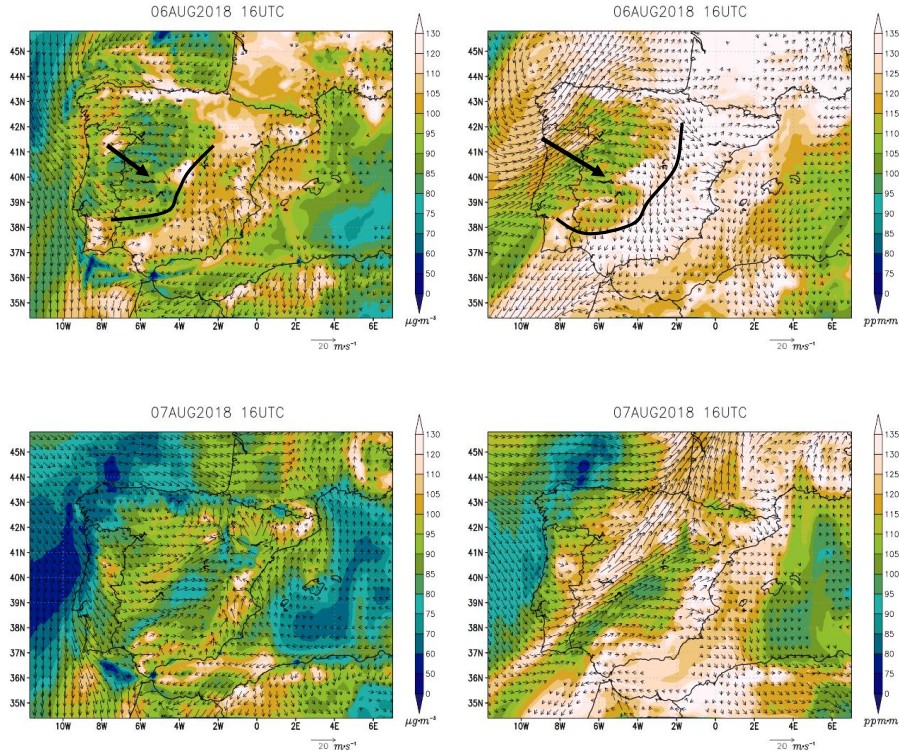

**Figure 16. Simulated $O_3$ concentrations (color scale) and wind fields (vectors) by WRF-CAMx in d02 at 16 UTC on August 6 and 7, 2018. Left panels show the ozone and wind concentration in surface and right panels the integrated ozone concentration up to 2500 m AGL and wind at 1250 m AGL. Winds lower than 2 m·s$^{-1}$ have been omitted.**





### 3.3.    Statistical evaluation of simulated $O_3$ concentrations

We have calculated the statistical metrics shown in Table 2 for 116 $O_3$ measurement stations (Figure 1) for the period from August to August 7. All these stations meet the criterion of data availability of more than 95% of hourly $O_3$ concentrations. Of the total number of stations, 83 are located in Spain, in NAI, and the remaining 33 are located in Portugal, in WAI. The averaged Pearson correlation coefficient (r) for all the stations was 0.58 and the Index Of Agreement (IOA) was 0.66. The CAMx model tends to overestimate $O_3$ concentrations for this region, with an averaged Mean Bias (MB) of +13.7 µg·m$^{-3}$ and an averaged Mean Error (ME) of 27.0 µg·m$^{-3}$. The calculated statistical parameters are within the range of values found in similar studies.

We have detected better model performance at WAI, where averaged r was 0.7 and IOA was 0.73, compared to 0.53 and 0.63, respectively, for the rest of the NAI stations. The same is true for the averaged MB values: -0.5 µg·m$^{-3}$ versus +19.4 µg·m$^{-3}$, and with an averaged ME of 25.2 µg·m$^{-3}$ versus 27.7 µg·m$^{-3}$ at WAI and NAI, respectively. This statistical difference could be due to an over-representation of some areas due to the proximity of measurement stations in NAI, and to the number of industrial stations that are exposed to industrial emission sources. Table S4 and Table S5 of the Supplementary material give detailed statistics for each of the stations.

It should be noted that the stations in NAI might lack representativeness for background $O_3$ measurements since many of them are located near industrial centers and there are few background stations that provide useful and reliable data to address $O_3$ transport and accumulation processes.

### 4.    Conclusions

This paper analyzes a tropospheric $O_3$ pollution episode that occurred over WAI and NAI, in Spain and Portugal, during August 2-6, 2018. The episode was characterized by an almost-simultaneous abrupt rise in $O_3$ concentrations in both regions, which remained high throughout the entire episode, exceeding the target values and the information threshold of the EC/50/2008 EU Directive. Using the meteorological and photochemical WRF-CAMx modeling system, we have identified the transport mechanisms behind this type of episodes, especially complex due to a meteorology characterized by a permanent wind shear throughout the entire period. Additionally, we have been able to characterize the possible sources of photochemical pollutants affecting these two areas.

The episode began with an accumulation of pollutants in the higher layers above 2,000 m AGL over IP, due to a decoupling of high altitude and surface air masses. The origin of that upper-level polluted air mass was probably due to the emission of pollutants during previous days in IP itself, which were then trapped in below 2,500 m ASL due to the stability of the upper warmer air. Subsequently, upper air masses fumigated onto the surface through the different orographic chimneys along the Atlantic coast, producing the beginning of the episode. During the initiation of the episode, the simulation pointed out that the dominant process was likely to be fumigation, with a contribution of 30-40 µg·m$^{-3}$ of the observed $O_3$ increase. Measured $O_3$ maximum daily concentrations increased in more than 40 µg·m$^{-3}$ from the previous day's highs, and simulated $O_3$ in more than 30 µg·m$^{-3}$.

From August 3 onwards, the fumigated air masses were joined by other polluted air masses. According to the simulation, NAI received $O_3$-polluted air masses imported from France, providing a minimum of 100 µg·m$^{-3}$ background $O_3$ concentrations, while WAI received $O_3$ from the N and center of IP, probably sharing the same minimum background contribution because of the continuity of those air masses. The most intense exceedances occurred in the sea-facing slopes of the main coastal ranges, at Valderejo (Basque Country, Spain) and Douro Norte (Portugal) measurement stations. In these sites, there is an additional impact of local coastal emissions to the already existing high background concentrations (100 µg·m$^{-3}$). That local contribution, introducing "fresh" pollutants inland with the sea breezes, produced concentrations above 130 µg·m$^{-3}$ of $O_3$ in the form of a peak of both measured and simulated concentrations, indicating a local



contribution of at least 30 µg·m⁻³ of $O_3$ in both locations. Those concentrations, as well as the transport pathways observed through the simulations, showed that during the episode there were different contributions and interrelated transport processes, first, an $O_3$ fumigation and interregional transport (within IP) during August 2, and then, from August 3 onwards, a continental European $O_3$ transport and concurrent accumulation within coastal circulations. The dissipation of the episode occurred gradually from W to E

due to an Atlantic advection, which introduced colder and cleaner air. After the front passed through, pollutants were carried from W to E, causing maximum hourly concentrations that were significant prior to the episode dissipation.

CAMx simulation conformed to the statistical parameters traditionally used with these models. We have introduced for the first time the analysis of winds in altitude and the calculation of integrated $O_3$

concentrations for a deeper understanding of this episode. These newly proposed analyses are necessary to understand air pollution episodes in areas with complex topography where re-circulatory processes can occur in both the upper and lower atmosphere. They allowed us to observe the medium and long-range transports of polluted air masses, abstracting from local effects, and, in turn, whether the increases in $O_3$ concentrations were due to air mass horizontal advections, fumigations, or a combination of both.

In view of the diversity of processes involved in this type of $O_3$ episodes, the authors of this article recommend extending the analysis of modeling studies to upper levels of the atmosphere, particularly in complex terrain applications and with complex meteorological situations such as this case. In order to improve predictions as well as control strategies, databases of observations should be expanded at the surface and upper levels of the atmosphere. WAI surface station measurements have proven to be

representative for evaluating the episode and agree with the simulations. Meanwhile, in NAI, measurement stations might not be very representative to address this kind of episodes due to their proximity to industrial sites. In upper levels of the atmosphere, $O_3$ soundings and LIDAR, among other techniques for the characterization of the vertical ozone distribution, should be used to further analyze the transport pathways and accumulation processes addressed in this paper.


**Author contributions**

Eduardo Torre-Pascual: Conceptualization, Data curation, Formal analysis, Investigation, Software, Visualization, Original draft preparation, Review & Editing.
Gotzon Gangoiti: Conceptualization, Data curation, Formal analysis, Investigation, Software,
Visualization, Original draft preparation, Review & Editing.
Ana Rodríguez-García: Conceptualization, Data curation, Formal analysis, Investigation, Software, Visualization, Original draft preparation, Review & Editing.
Estibaliz Sáez de Cámara: Conceptualization, Data curation, Investigation, Software, Visualization, Original draft preparation, Review & Editing.
Joana Ferreira: Conceptualization, Data curation, Investigation, Visualization, Original draft preparation, Review & Editing.
Carla Gama: Conceptualization, Data curation, Investigation, Visualization, Original draft preparation, Review & Editing.
María Carmen Gómez: Conceptualization, Investigation, Visualization, Original draft preparation,
Review & Editing.
Iñaki Zuazo: Conceptualization, Data curation, Investigation, Visualization, Original draft preparation, Review & Editing.
Jose Antonio García: Conceptualization, Investigation, Visualization, Original draft preparation, Review & Editing.
Maite de Blas: Conceptualization, Investigation, Visualization, Original draft preparation, Review & Editing.



**Competing interests**

The authors declare that they have no conflict of interest.

**Acknowledgments**

The authors wish to thank the Department of Economic Development, Sustainability and the Environment
of the Basque Government (CONV 22/16) and the University of the Basque Country UPV/EHU as the
source of our main financial support: GIA consolidated Research Groups (GIU21/050)
(https://www.ehu.eus/es/web/gia). In addition to the Spanish Ministry of Universities and the European
Union, for the Margarita Salas Grant (MARSA21/23) of Eduardo Torre-Pascual, funded by the European
Union-Next Generation EU. These financing bodies have played an exclusively economic role in the study.

Thanks are also due for the financial support to the contract grants of J. Ferreira (2020.00622.CEECIND)
and C. Gama (2021.00732.CEECIND), to CESAM (UIDP/50017/2020+ UIDB/50017/2020+
LA/P/0094/2020), to FCT/MCTES through national funds and the co-funding by the FEDER, within the
PT2020 Partnership Agreement and Compete 2020.

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
