# Peer review of "Analysis of an intense O3 pollution episode in the Atlantic Coast of the Iberian Peninsula using photochemical modeling: characterization of transport pathways and accumulation processes."

_EGUsphere, 2023_

## Author Comment (AC1)

**Analysis of an intense $O_3$ pollution episode in the Atlantic Coast of the Iberian Peninsula using photochemical modeling: characterization of transport pathways and accumulation processes.**

Eduardo Torre-Pascual[1], Gotzon Gangoiti[1], Ana Rodríguez-García[1], Estibaliz Sáez de Cámara[1], Joana Ferreira[2], Carla Gama[2], María Carmen Gómez[1], Iñaki Zuazo[1], Jose Antonio García[1], Maite de Blas[1].

[1]Faculty of Engineering Bilbao, University of the Basque Country (UPV/EHU), Bilbao, 48013, Spain

[2]CESAM & Department of Environment and Planning, University of Aveiro, Aveiro, 3810-193, Portugal

*Correspondence to:* Eduardo Torre-Pascual (eduardo.delatorre@ehu.eus)

**Reply on RC1**

We want to thank the referee for his/her dedication, time, and thorough examination of our manuscript. We have followed the suggestions provided to us to enhance the manuscript in accordance with the recommendations.

**Line 15: has a similar approach, i.e., combing upper and lower tropospheric parameters, been used to investigate tropospheric O3 in other studies?**

As far as the authors are aware, studies employing the specific methodology outlined in this manuscript have not been conducted. While comparisons of vertical ozone profiles using modeling tools and measurements, and vertical distribution comparison via LIDAR have been performed, there is no analysis that combines surface and upper-level data simultaneously to showcase how $O_3$ phenomena occurring at higher layers can influence surface conditions. This limitation primarily stems from the fact that vertical and horizontal recirculations occur in areas of complex orography that have been scarcely examined, except for the Mediterranean region, as mentioned in the introduction.

**Lines 25-26: need to be rephrased.**

**Lines 30-33: the authors mention the lack of detailed research on $O_3$ issues in the Atlantic Coast of the Iberian Peninsula but do not explain why studying this region is important. It would be beneficial to include a sentence to explain the significance and value of investigating $O_3$ problems in this area.**

Studying ozone in the Atlantic Coast of the Iberian Peninsula is important as this region is consistently exposed to very high levels of ozone. For instance, in 2020, Portugal, situated on the Atlantic Coast, registered maximum $O_3$ levels among European regions, as mentioned earlier in line 27. These high $O_3$ concentrations, which surpass the standards defined for the protection of human health by the European Directive on ambient air quality and cleaner air, are due to a combination between the northern mid-latitudes background concentrations (Cuevas et al., 2013; Rodrigues et al., 2021) and the local and

regional production favored by the region's circulation weather patterns (Russo et al., 2016) and meteorological conditions, including temperature (Sá et al., 2015) and solar radiation (Silva and Pires, 2022). Additionally, on the Spanish Atlantic coast, high ozone concentration episodes are also measured. Despite being the region least affected in Spain and having a lower frequency of this type of pollution, regions like the Basque Country have intermediate ozone concentrations and are impacted by high stable background levels. Ozone reaching this area, coupled with local emission sources, can also have downstream repercussions, such as in the Duero and Ebro Valley in Spain, and Portugal. Specifically, the station examined in greater detail in this manuscript, Valderejo, holds special significance as it consistently surpasses the mentioned standards. This station is situated in a convergence zone of different air masses, providing useful insights about regional ozone transport and it is often analyzed within the Spanish National Ozone Plan framework.

Considering the more stringent values of the World Health Organization (WHO) air quality guidelines (which are planned to be incorporated into the revised air quality directive by 2035), the situation becomes even more concerning.

The authors have revised the original paragraph to convey this rationale.

Suggestion for the final text in the manuscript (changes highlighted in red):

Southern European countries are heavily exposed to high tropospheric ozone ($O_3$) concentrations, particularly those surrounding the Mediterranean Basin (ETC/ACM, 2018; EEA, 2019). The 2020 European air quality report indicated a decrease in $O_3$ levels compared to previous years. However, levels remained notably high, with maximum concentrations observed in central Europe, certain Mediterranean countries, and Portugal - located on the Atlantic Coast of the Iberian Peninsula (IP) (EEA, 2022). These high $O_3$ concentrations are due to a combination between the northern mid-latitudes background concentrations (Cuevas et al., 2013; Rodrigues et al., 2021) and the local and regional production favored by the region's circulation weather patterns (Russo et al., 2016) and meteorological conditions, including temperature (Sá et al., 2015) and solar radiation (Silva and Pires, 2022). However, despite their importance, $O_3$ episodes in the Atlantic Coast, specifically in Northern Atlantic Iberia (NAI) and Western Atlantic Iberia (WAI), have not been examined in detail. In this region, significant episodes of tropospheric $O_3$ have occurred, with values exceeding the target value defined for the protection of human health defined by the Directive 2008/50/EC (Silva and Pires, 2022). Moreover, considering the more stringent values of the World Health Organization (WHO) air quality guidelines (which are planned to be incorporated into the revised air quality directive by 2035), the situation becomes even more concerning.

New references:

Cuevas, E., González, Y., Rodríguez, S., Guerra, J. C., Gómez-Peláez, A. J., Alonso-Pérez, S., Bustos, J., and Milford, C.: Assessment of atmospheric processes driving ozone variations in the subtropical North Atlantic free troposphere, Atmos. Chem. Phys., 13, 1973–1998, https://doi.org/10.5194/acp-13-1973-2013, 2013.

Rodrigues, V., Gama, C., Ascenso, A., Oliveira, K., Coelho, S., Monteiro, A., Hayes, E., and Lopes, M.: Assessing air pollution in European cities to support a citizen centered approach to air quality management, Sci. Total Environ., 799, 149311, https://doi.org/10.1016/j.scitotenv.2021.149311, 2021.

Russo, A., Gouveia, C., Levy, I., Dayan, U., Jerez, S., Mendes, M., and Trigo, R.: Coastal recirculation potential affecting air pollutants in Portugal: The role of circulation weather types, Atmos. Environ., 135, 9–19, https://doi.org/10.1016/j.atmosenv.2016.03.039, 2016.

Sá, E., Tchepel, O., Carvalho, A., and Borrego, C.: Meteorological driven changes on air quality over Portugal: A KZ filter application. Atmos. Pol. Res., 6(6), 979–989, https://doi.org/10.1016/j.apr.2015.05.003, 2015.

Silva, R.C.V. and Pires, J.C.M.: Surface Ozone Pollution: Trends, Meteorological Influences, and Chemical Precursors in Portugal, Sustainability, 14(4), 2383, https://doi.org/10.3390/su14042383, 2022.

**Lines 64-65: it would be better to explain the selection of the O3 pollution episode in Aug 2-6 2018.**

We have rephrased lines 64-66 for a better understanding of the episode selection:

Suggestion for the final text in the manuscript (changes highlighted in red):

We have selected an $O_3$ pollution episode lasting five days occurring from August $2^{nd}$ to $6^{th}$, 2018 (see Section 3.3), which affected Spain and Portugal. This episode was characterized by a notable and simultaneous increase in $O_3$ concentration levels across both the NAI and WAI regions during August 2, high $O_3$ concentrations during consecutive days until August $6^{th}$, and final dissipation on August $7^{th}$.

**Section 3.1 focuses on the validation of meteorological variables, which is not the major target of this work. It could be better to move some less important figures and tables in this section to the supplement.**

We have moved figures 4, 5, 7, 8, 9 and 10 to supplementary and retaining figures 3 (synoptic evolution from wetterzentrale) and 6 (radiosonde data and WRF comparison) in the manuscript.

**Figures 13-16: panel numbers (i.e., a, b, c) should be added and mentioned in the discussion to facilitate the audience. It could be helpful to integrate Figures 13 and 16 to show O3 concentration evolution in 1-7 August continuously instead of 1-3 and 6-7 August in two figures. Furthermore, the discussions in this section are difficult to follow, particularly, the discussions about Figures 14 and 15.**

We have added panel numbers accordingly. In the revised manuscript the already referenced figures will be linked to the specific graph in each panel. We prefer to separate the beginning of the episode and the end of the episode because the explanations for the beginning are far apart in the text from the ones of the dissipation.

Regarding the discussions about Figures 14 and 15 we suggest the following modifications:

concentrations above 110 μg·m⁻³ (55 ppb in Figure 14). Above 500 m ASL horizontally projected winds were from the S, while below 500 m ASL sea breezes prevailed near the coast, and O₃ concentrations were not significantly high in coastal areas (40 ppb). In the afternoon of August 2, horizontally projected winds above 1,000 m ASL turned to the N and began withdrawing the polluted air mass towards the S of IP. Therefore, the first peak recorded in Valderejo on day 2 (Figure 12) was due to that fumigation process.

In VNP, on August 1, the higher polluted air mass was positioned above 1,800 meters above sea level (ASL) as indicated by the simulated O₃ concentrations (Figure 14 **a**. and **b**.). On August 2, starting from 12 UTC, this high-altitude polluted air mass was mixed with surface air masses in the inland valleys. This led to simulated O₃ concentrations exceeding 110 μg·m⁻³ (55 ppb as shown in Figure 14 **c**). Winds at a horizontal projection above 500 m ASL were from S, while below 500 m ASL sea breezes from N prevailed near the coast, resulting in O₃ concentrations not significantly high along the coastal areas (40 ppb). By the afternoon of August 2 (Figure 14 **d**), winds at a horizontal projection above 1,000 m ASL shifted to the N, causing the polluted air mass to recede towards the south of the IP. Consequently, the initial peak observed in Valderejo on day 2 (as depicted in Figure 12) was a result of this fumigation process. From August 3 onwards (Figure 14 **e** and **f**), we observed the influx of O₃ polluted air masses of European origin below 1,500 m ASL transported across the sea.

In DN, the upper-level polluted air mass entered at approximately 1,800 m ASL during August 1, with S-SE origin (Figure 15). Low O₃ concentrations were observed over the sea surface, while on the coastline there was a slight increase in concentrations in the lower layers between 400 and 1,000 m ASL due to the transport of locally emitted pollutants towards the inland areas pushed by sea breezes with northwesterly winds. In the inland valleys, high O₃ concentrations were registered due to the fumigation of the polluted air masses at altitude, due to convective movements during the afternoon (Figure 15). The coastal return flows during the afternoon, at 900 m ASL (Figure 15), could transport part of that fumigated O₃ to the sea surface, generating an O₃ reservoir for the following days, similar to the processes described for Mediterranean areas (Millán et al., 2002).

In DN, on August 1, an upper-level polluted air mass entered at approximately 1,800 meters above sea level (ASL) originated in the S/SW (Figure 15 **a**). While low O₃ concentrations were observed over the sea surface, a slight increase in concentrations occurred along the coastline in the lower atmospheric layers between 400 and 1,000 meters ASL. Inland valleys experienced high O₃ concentrations due to the pollution carried at higher altitudes and mixed into the surface, propelled by convective movements during the afternoon (Figure 15 **b**). Return flows along the coast at 900 meters ASL during the afternoon (Figure 15 **b**) potentially carried some of this elevated O₃ back to the sea surface, creating an O₃ reservoir for subsequent days. This mechanism resembles processes described for Mediterranean regions (Millán et al., 2002), resulting in sea-land recirculations over several consecutive days, forming injections of ozone into upper layers that may return to the coast during the following days (Figure 15 **c** to **f**).

**Section 3.3: It is suggested to move the O₃ validation in this section forward because reliable simulation of O₃ concentrations is the prerequisite for the simulation-based analysis shown in Section 3.2. Figures and tables are required to be shown to demonstrate the validation results.**

We agree with the reviewer's comment about moving the section. In the revised version of the manuscript, the validation section appears before presenting the simulation-based analysis (see the new version with changes highlighted in red).

Regarding the validation results, Tables S4 and S5 (part of Supplement) already presented the detailed statistics for each of the 116 air quality stations. Now, in the revised version of the manuscript, we have also included figures showing the dispersion and spatial distribution of values for each metric. With the new figures, we improved the discussion on modelling performance, highlighting which stations have the best and the worse performance, while discussing reasons for that. We also included a time series plot of modelled and observed $O_3$ spatially average concentrations, showing the good performance of the model in reproducing the $O_3$ daily patterns, overestimating however the lowest concentrations observed during the night during this episode. Overall, in the new version of the manuscript, we have improved the way we show that the model is suitable for our purpose.

Suggestion for the final text in the manuscript (changes highlighted in red):

We have calculated the statistical metrics shown in Table 2 for 116 $O_3$ measurement stations (Figure R1, to be included in the supplement) for the period from August 1 to 7 (we are removing RMSE from Table 2). All these stations meet the criterion of data availability of more than 95% of hourly $O_3$ concentrations. Of the total number of stations, 83 are located in Spain, in NAI, and the remaining 33 are located in Portugal, in WAI. The dispersion of the individual metrics is shown in Figure R1. The median Pearson correlation coefficient (r) for all the stations was 0.66 and the median Index of Agreement (IOA) was 0.70. The CAMx model tends to overestimate $O_3$ concentrations for this region, as shown by the box and whisker plots of the Mean Bias (MB): the interquartile values, from the 25th percentile to the 75th percentile, are all positive. The median Mean Bias (MB) is +14.0 $\mu g \cdot m^{-3}$ and the median Mean Error (ME) is 24.0 $\mu g \cdot m^{-3}$. The calculated statistical parameters are within the range of values found in similar studies.

[Figure]

Figure R1. Box and whisker plots for the statistical metrics (MB, ME, IOA, and r) calculated for the 116 $O_3$ measurement stations. The boxes span from the first to the third quartile and the line inside the boxes represents the median. Whiskers extend to the most extreme data points within 1.5 times the interquartile range. Any metrics beyond this range are considered outliers and are plotted as individual points (+).

Statistical metrics calculated for each site are represented in Figure R2 (specific values are being updated in Table S4 and Table S5 of the Supplementary material). We have detected better model performance at WAI, where median r was 0.74 and IOA was 0.79, compared to 0.62 and 0.66, respectively, for the rest of the NAI stations. The same is true for the median MB values: -1.0 $\mu g \cdot m^{-3}$ versus +20.0 $\mu g \cdot m^{-3}$, and with a median ME of 20.0 $\mu g \cdot m^{-3}$ versus 26.0 $\mu g \cdot m^{-3}$ at WAI and NAI, respectively. This statistical difference could be due to an over-representation of some areas due to the

proximity of measurement stations in NAI, and to the number of industrial stations that are exposed to industrial emission sources.

In WAI, three stations are highlighted as having a poor model performance: the urban background PT01044 (Paços de Ferreira, Porto) in the North, and the suburban industrial PT04001 (Monte Chãos) and rural background PT04002 (Monte Velho), in the southwest of Portugal. Paços de Ferreira municipality stands out due to its furniture and textile industry. The largest positive Mean Bias error (+68 µg·m⁻³) calculated for this area indicates that the model is strongly overestimating O₃ concentrations, which may be due to unrealistic NOₓ emissions such as lack of local NO emissions in the model, affecting the modelled O₃ concentrations through the underestimation of the NOₓ titration process. Although Monte Chãos and Monte Velho are both located near the Sines Industrial and Logistics Zone, the largest industrial area in Portugal, other factors than industrial emissions may be playing a crucial role in the modelling performance: a large forest fire took place in Monchique, from the 3ʳᵈ to the 10ᵗʰ of August, burning around 27,000 hectares of forest and agricultural land, emitting a huge amount of pollutants, and thus affecting air quality. On the contrary, the rural background PT01048 (Douro Norte), in the North, exhibits the best statistical metrics, with IOA=0.88 and ME= 10 µg·m⁻³.

In NAI, the rural background ES1599A (Pagoeta) and the urban background ES1747A (Rotxapea), both located at the Eastern part of NAI, have the best performance, with IOA= 0.86 and 0.83, respectively.  The overall behavior of the statistical data in this area exhibits a strong correlation, with high IOA and r values, albeit with a general overestimation of O₃ levels. Stations in NAI might lack representativeness for background O₃ measurements since many of them are located near industrial centers with high NOₓ emissions and there are few background stations that provide useful and reliable data to address O₃ transport and accumulation processes.

[Figure]

Figure R2. Spatial distribution of the values of the Mean Bias (MB), Mean Error (ME), Index of Agreement (IOA), and Pearson correlation coefficient (r).

For every hourly interval, we computed the average $O_3$ concentrations of observed and simulated data across all 116 stations within the domain. This allowed us to determine the overall average $O_3$ concentration across all sites in the domain (Figure R3). While this process involved pairing data temporally, it did not differentiate spatial distribution. The graphical representation demonstrates that WRF-CAMx generally replicates the daily $O_3$ patterns well and overestimates the lowest observed concentrations, particularly during nighttime in this specific episode. Our assessment reveals the model's ability to capture the initial sudden rise in $O_3$ concentrations, both of which depict an increase of approximately +25-30 $\mu g \cdot m^{-3}$ in maximum concentrations compared to the preceding day on August 2. During the following days, maximum values persist consistently above 120 $\mu g \cdot m^{-3}$, indicating persistent elevated average levels. The decline observed on August 7 is also well replicated along with the dissipation of the episode. Despite nighttime discrepancies, our evaluation suggests that the model's application remains suitable for the objectives of our research.

[Figure]

Figure R3. Time series plot of modelled and observed $O_3$ average concentrations in the 116 sites considered, between August 1 and 7, 2018 (average concentrations considering pairing in time but not pairing in space).

---

## Author Comment (AC2)

**Analysis of an intense O₃ pollution episode in the Atlantic Coast of the Iberian Peninsula using photochemical modeling: characterization of transport pathways and accumulation processes.**

Eduardo Torre-Pascual[1], Gotzon Gangoiti[1], Ana Rodríguez-García[1], Estibaliz Sáez de Cámara[1], Joana Ferreira[2], Carla Gama[2], María Carmen Gómez[1], Iñaki Zuazo[1], Jose Antonio García[1], Maite de Blas[1].

[1]Faculty of Engineering Bilbao, University of the Basque Country (UPV/EHU), Bilbao, 48013, Spain

[2]CESAM & Department of Environment and Planning, University of Aveiro, Aveiro, 3810-193, Portugal

*Correspondence to:* Eduardo Torre-Pascual (eduardo.delatorre@ehu.eus)

**Reply on RC2**

We want to thank the referee for his/her dedication, time, and thorough examination of our manuscript. We have followed the suggestions provided to us to enhance the manuscript in accordance with the recommendations.

**line 21: heading --> accompanying ?**

Thank you for your suggestion, we are modifying it properly.

**line 70: I'm not sure what "pathway in the Atlantic axis of the IP" means**

We wanted to express the northern region of the Iberian Peninsula in the Atlantic Axis, not in the Mediterranean one. We have rephrased that line with the following (changes highlighted in red):

Valdenebro et al. (2011) proposed the existence of a potential transport pathway for O₃ and pollutants along the Atlantic axis of the Iberian Peninsula, in NAI.

**Figure 5: ERA5 surface and 750 hPa winds for August 6 and 7 is not called in the text**

Thank you for noticing this detail. We are calling them in the text.

**Section 3.1.2:  Should point out that there is a greater observed amplitude of the daily cycle of wind speed at both Spanish stations than in the model.  But it is interesting that the reverse is true for the stations in Portugal.**

We have added the following at the end of the first paragraph:

We have observed a more pronounced diurnal variability in wind speed at both Spanish stations compared to the model, suggestive of the influence of this local breeze phenomena. The intensity difference may affect the extent of the emitted $O_3$ precursors' dispersion.

**line 492:   ppm-m is a rather strange unit.  How is this calculated?  Why not use something more customary for vertical columns, such as molecules per cm\*\*2 ?    Figures 13 and 16 look like they use this unit also.**

We selected this measurement method because it doesn't encompass the entire ozone column but rather aims to swiftly estimate the 'average' concentration across a specific atmospheric thickness. We focused on an atmospheric with a thickness of 2,500 meters above sea level, where we anticipate the significance of higher atmospheric layers. Observations in Portugal demonstrate the potential mixing of air masses between surface and higher altitudes. Our calculation involved multiplying the simulated ozone concentration of each layer by its thickness. For example, if two layers exhibit an identical concentration of 0.060 parts per million (ppm) and have a thickness of 20 meters each, the integrated ozone for 40 meters above sea level would be obtained by multiplying 0.060x20 + 0.060x20. This unit becomes useful when dividing ppm by the total integrated height, offering a rapid estimation of the 'average' concentration throughout that particular atmospheric thickness. However, it's important to note the uncertainty as the higher layers progressively increase in altitude, and the concentrations of thicker layers might carry more weight in the overall assessment.

**line 501: ...O3 concentrations to values exceeding....**

Now corrected.

**line 605:  ...from August 2 to August 7.**

"2" was missing, now corrected.

---

## Author Response (AR2)

**Response to the editor**

We have completed the minor changes as requested by the referees and, furthermore, conducted a final review to rectify any typographical errors. Should you require any further assistance, please do not hesitate to contact us. We sincerely appreciate your expeditious handling of the matter.

Kind regards,

Eduardo Torre-Pascual